# Extremely Simple Multimodal Outlier Synthesis for Out-of-Distribution Detection and Segmentation

**Moru Liu**[1*]  **Hao Dong**[2*]  **Jessica Kelly**[3]  **Olga Fink**[4]  **Mario Trapp**[1,3]

[1]Technical University of Munich  [2]ETH Zürich  [3]Fraunhofer IKS  [4]EPFL

## Abstract

Out-of-distribution (OOD) detection and segmentation are crucial for deploying machine learning models in safety-critical applications such as autonomous driving and robot-assisted surgery. While prior research has primarily focused on unimodal image data, real-world applications are inherently multimodal, requiring the integration of multiple modalities for improved OOD detection. A key challenge is the lack of supervision signals from unknown data, leading to overconfident predictions on OOD samples. To address this challenge, we propose **Feature Mixing**, an extremely simple and fast method for multimodal outlier synthesis with theoretical support, which can be further optimized to help the model better distinguish between in-distribution (ID) and OOD data. Feature Mixing is modality-agnostic and applicable to various modality combinations. Additionally, we introduce CARLA-OOD, a novel multimodal dataset for OOD segmentation, featuring synthetic OOD objects across diverse scenes and weather conditions. Extensive experiments on SemanticKITTI, nuScenes, CARLA-OOD datasets, and the MultiOOD benchmark demonstrate that Feature Mixing achieves state-of-the-art performance with a $10\times$ to $370\times$ speedup. Our source code and dataset will be available at https://github.com/mona4399/FeatureMixing.

## 1 Introduction

Classification and segmentation are fundamental computer vision tasks that have seen significant advancements with deep neural networks [22, 40]. However, most models operate under a closed-set assumption, expecting identical class distributions in training and testing. In real-world applications, this assumption often fails, as out-of-distribution (OOD) objects frequently appear. Ignoring OOD instances poses critical safety risks in domains like autonomous driving and robot-assisted surgery, motivating research on OOD detection [36] and segmentation [7] to identify *unknown* objects that are unseen during training.

Most existing OOD detection and segmentation methods focus on unimodal inputs, such as images [36] or point clouds [7], despite the inherently multimodal nature of real-world applications. Leveraging multiple modalities can provide complementary information to improve performance [15, 14]. Recent work by Dong et al. [17] introduced the first multimodal OOD detection benchmark and framework and also extended the framework to the multimodal OOD segmentation task. A key challenge in OOD detection and segmentation is the tendency of neural networks to assign high confidence scores to OOD inputs [44] due to the lack of explicit supervision for unknowns during training. While real outlier datasets [25] can help mitigate this, they are often costly and impractical to obtain. Alternatively, synthetic outliers [19, 48, 43] have been proven effective for regularization, but existing methods are designed for unimodal scenarios and struggle in multimodal settings [17]. Dong

---

*Equal contribution.

39th Conference on Neural Information Processing Systems (NeurIPS 2025).

et al. [17] proposed a multimodal outlier synthesis technique using nearest-neighbor information, but its computational cost remains prohibitive for segmentation tasks.

To address this, we propose ***Feature Mixing***, an extremely simple and efficient multimodal outlier synthesis method with theoretical support. Given in-distribution (ID) features from two modalities, Feature Mixing randomly swaps a subset of $N$ feature dimensions between them to generate new multimodal outliers. By maximizing the entropy of these outliers during training, our method effectively reduces overconfidence and enhances the model's ability to distinguish OOD from ID samples. Feature Mixing is modality-agnostic and applicable to various modality combinations, such as images and point clouds or video and optical flow. Moreover, its lightweight design enables a $10\times$ speedup for multimodal OOD detection and a $370\times$ speedup for segmentation compared to [17] (Fig. 1).

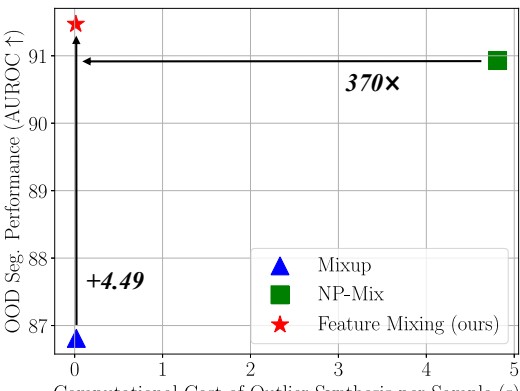

We conduct extensive evaluations across *eight datasets* and *four modalities* to validate the effectiveness of Feature Mixing. For multimodal OOD detection, we use five datasets from the MultiOOD benchmark [17] with video and optical flow modalities. For multimodal OOD segmentation, we evaluate on large-scale real-world datasets, including SemanticKITTI [3] and nuScenes [6], with image and point cloud modalities. To address the lack of multimodal OOD segmentation datasets, we introduce ***CARLA-OOD***, a synthetic dataset generated using CARLA simulator [18], featuring diverse OOD objects in various challenging scenes and weather conditions (Fig. 6). Our experiments on both synthetic and real-world datasets demonstrate that Feature Mixing outperforms existing outlier synthesis methods in most cases with a significant speedup. In summary, the main contributions of this paper are:

Figure 1: Mixup [52] is efficient for outlier synthesis but performs poorly in OOD segmentation. In contrast, NP-Mix [17] achieves strong OOD segmentation but is computationally expensive. Our Feature Mixing combines both speed and performance, benefiting from its simple yet effective design. Results are on SemanticKITTI dataset.

1. We introduce Feature Mixing, an extremely simple and fast method for multimodal outlier synthesis, applicable to diverse modality combinations.

2. We provide theoretical insights in support of the efficacy of Feature Mixing.

3. We present the challenging CARLA-OOD dataset with diverse scenes and weather conditions, addressing the scarcity of multimodal OOD segmentation datasets.

4. We conduct extensive experiments across eight datasets and four modalities to demonstrate the effectiveness of our proposed approach.

## 2 Related Work

### 2.1 Out-of-Distribution Detection

OOD detection aims to detect test samples with semantic shift without losing the ID classification accuracy. Numerous OOD detection algorithms have been developed. Post hoc methods [24, 23, 36] aim to design OOD scores based on the classification output of neural networks, offering the advantage of being easy to use without modifying the training procedure and objective. Methods like Mahalanobis [32] and $k$-nearest neighbor [47] use distance metrics in feature space for OOD detection, while virtual-logit matching [50] integrates information from both feature and logit spaces to define the OOD score. Additionally, some approaches propose to synthesize outliers [19, 48] or normalize logits [51] to address prediction overconfidence by training-time regularization. However, all these approaches are designed for unimodal scenarios without accounting for the complementary nature of multiple modalities.

## 2.2 Out-of-Distribution Segmentation

OOD segmentation focuses on pixel- or point-level segmentation of OOD objects and has been widely studied in medical images [1], industrial inspection [46], and autonomous driving [5] in recent years. Approaches to pixel-level segmentation are generally categorized into uncertainty-based [27, 49], outlier exposure [38, 31], and reconstruction-based methods [2, 42]. Point-level segmentation has gained attention in recent years due to its practical applications in real-world environments. For example, Cen et al. [7] address OOD segmentation on LiDAR point cloud and train redundancy classifiers to segment unknown object points by simulating outliers through the random resizing of known classes. Li et al. [33] separate ID and OOD features using a prototype-based clustering approach and employing a generative adversarial network [21] to synthesize outlier features. Similarly, these techniques are exclusively focused on unimodal contexts, neglecting the inherent complementarity among different modalities.

## 2.3 Multimodal OOD Detection and Segmentation

Multimodal OOD detection and segmentation are emerging research areas with limited prior work. Dong et al. [17] introduced the first multimodal OOD detection benchmark, identifying modality prediction discrepancy as a key indicator of OOD performance. They proposed the agree-to-disagree algorithm to amplify this discrepancy during training and developed a multimodal outlier synthesis method that expands the feature space using nearest-neighbor class information. Their approach was later extended to multimodal OOD segmentation on SemanticKITTI [3]. More recently, Li et al. [34] introduced dynamic prototype updating, which adjusts class centers to account for intra-class variability in multimodal OOD detection. In this work, we propose a novel multimodal outlier synthesis method applicable to both OOD detection and segmentation tasks.

## 3 Methodology

### 3.1 Problem Setup

In this work, we focus on multimodal OOD detection and segmentation, where multiple modalities are involved to help the model better identify unknown objects. We define the problem setups for each task below.

**Multimodal OOD Segmentation** aims to accurately segment both ID and OOD objects in a point cloud using LiDAR and image data. Given a training set with classes $\mathcal{Y} = \{1, 2, ..., C\}$, unlike traditional closed-set segmentation where test classes match training classes, OOD segmentation introduces unknown classes $\mathcal{U} = \{C + 1\}$ in the test set. A paired LiDAR point cloud and RGB image can be represented as $\mathcal{D} = \{\mathbf{P}, \mathbf{X}, \mathbf{y}\}$, where $\mathbf{P} = \{\mathbf{p}_1, \mathbf{p}_2, ..., \mathbf{p}_M\}$ denotes the LiDAR point cloud consisting of $M$ points, with each point $\mathbf{p}$ represented by three coordinates and intensity $\mathbf{p} = (x, y, z, i)$. Let $\mathbf{X} \in \mathbb{R}^{3 \times H \times W}$ represent the RGB image, where $H$ and $W$ denote height and width. The label $\mathbf{y} = \{y_1, y_2, ..., y_M\}$ provides semantic labels for each point, where $y \in \mathcal{Y}$ for the training data and $y \in \mathcal{Y} \cup \mathcal{U}$ for the test data.

Given a model $\mathcal{M}$ trained under the closed-set assumption, with its outputs $\mathbf{O} = \mathcal{M}(\mathbf{P}, \mathbf{X}) \in \mathbb{R}^{M \times C}$ within the domain of $\mathcal{Y}$. During deployment, $\mathcal{M}$ should accurately classify known samples in $\mathcal{Y}$ as ID and identify *unknown* samples in $\mathcal{U}$ as OOD. A separate score function $S(\mathbf{p})$ is typically used as an OOD module to decide whether a sample point $\mathbf{p} \in \mathbf{P}$ is from ID or OOD:

$$G_\eta(\mathbf{p}) = \begin{cases} \text{ID} & S(\mathbf{p}) \geq \eta \\ \text{OOD} & S(\mathbf{p}) < \eta \end{cases}, \tag{1}$$

where samples with higher scores $S(\mathbf{p})$ are classified as ID and vice versa, and $\eta$ is the threshold.

**Multimodal OOD Detection** aims to identify samples with semantic shifts in the test set using video and optical flow, where unknown classes are introduced. The setup is similar to segmentation but differs in input and output types. The input consists of a paired video $\mathbf{V}$ and optical flow $\mathbf{F}$, represented as $\mathcal{D} = \{\mathbf{V}, \mathbf{F}, y\}$, where $y$ is the sample-level label rather than point-level label in segmentation. Similarly, the model produces sample-level outputs $\mathbf{O} = \mathcal{M}(\mathbf{V}, \mathbf{F}) \in \mathbb{R}^C$ instead of point-level. The remaining setup follows that of segmentation, and we refer the reader to [17] for a detailed definition.

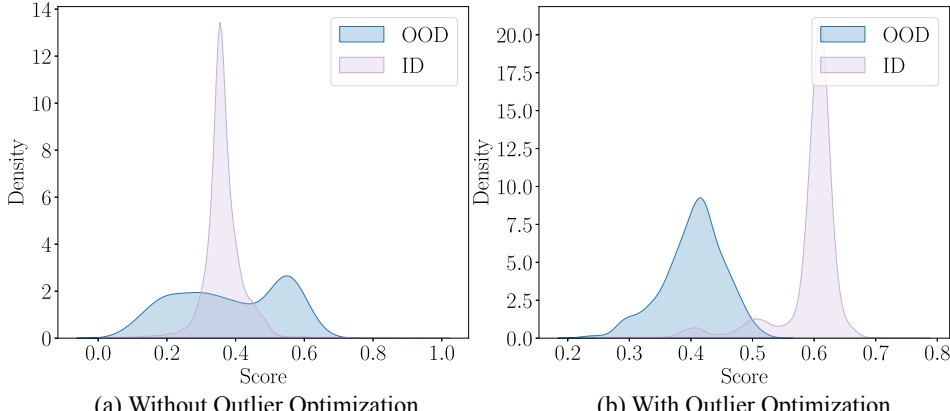

(a) Without Outlier Optimization       (b) With Outlier Optimization

Figure 2: (a) Uncertainty-based OOD methods face overconfidence issues, resulting in significant overlap between the score distributions of ID and OOD samples. (b) After training with outlier optimization, the confidence scores for ID and OOD samples become more distinct, enabling the model to better differentiate them. Results are on CARLA-OOD dataset.

## 3.2 Motivation for Outlier Synthesis

Uncertainty-based OOD detection and segmentation methods [24, 36, 37] are computationally efficient but suffer from overconfidence issues, as illustrated in Fig. 2 (a). Outlier exposure-based methods [25, 38, 31] mitigate this issue by training models using auxiliary OOD datasets to calibrate the confidences of both ID and OOD samples. However, such datasets are often unavailable, especially in multimodal settings. To address this challenge, we introduce Feature Mixing, an extremely simple and efficient multimodal outlier synthesis method that operates in the feature space with negligible computational overhead (Sec. 3.3). These synthesized outliers are further optimized via entropy maximization, enhancing the model's ability to distinguish ID from OOD data (Sec. 3.4). As shown in Fig. 2 (b), training with outlier optimization results in well-separated confidence scores, leading to improved OOD detection and segmentation.

## 3.3 Feature Mixing for Multimodal Outlier Synthesis

**Existing Methods.** Some prior works [49, 8] generate outliers in the pixel space by extracting OOD objects from external datasets and pasting them into inlier images. However, such methods are impractical for multimodal scenarios, where we need to generate outliers for paired multimodal data. Instead, generating outliers in the feature space is more effective and scalable. Mixup [52] interpolates features of randomly selected samples to generate outliers but inadvertently introduces noise samples within the ID distribution (Fig. 4 (a)). VOS [19] samples outliers from low-likelihood regions of the class-conditional feature distribution but is designed for unimodal settings and struggles with multimodal data. Moreover, it generates outliers too close to ID samples (Fig. 4 (b)) and is slow for high-dimensional features. NP-Mix [17] explores broader embedding spaces using nearest-neighbor class information but remains computationally expensive for segmentation tasks and introduces unwanted noise (Fig. 4 (c)).

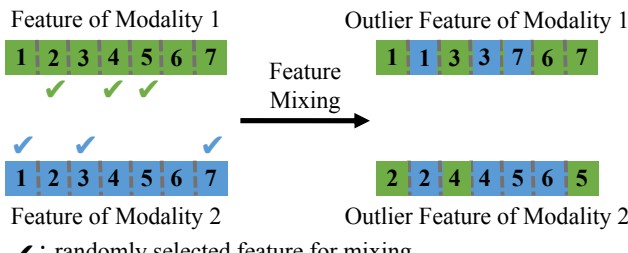

Figure 3: Illustration of Feature Mixing.

**Our Solution.** To overcome these limitations, we propose Feature Mixing, an extremely simple yet effective approach that generates multimodal outliers directly in the feature space. Our method ensures that the synthesized features remain distinct from ID features (Theorem 1) while preserving semantic consistency (Theorem 2). Given ID features $\mathbf{F} = [\mathbf{F}_c; \mathbf{F}_l]$, where $\mathbf{F}_c$ is from modality 1 and $\mathbf{F}_l$ is from modality 2, Feature Mixing randomly selects a subset of $N$ feature dimensions from each modality and swaps them to obtain new features $\widetilde{\mathbf{F}}_c$ and $\widetilde{\mathbf{F}}_l$, which are then concatenated to form the

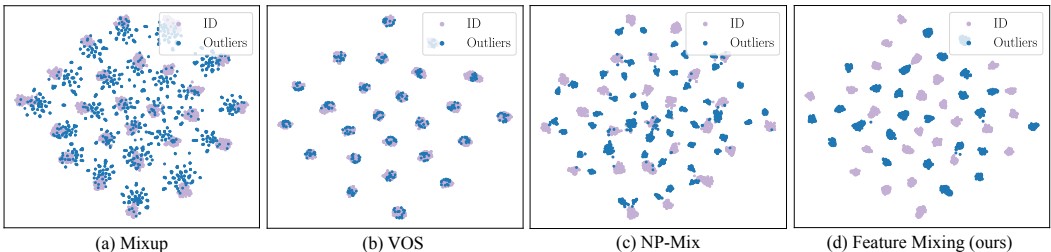

| (a) Mixup | (b) VOS | (c) NP-Mix | (d) Feature Mixing (ours) |

Figure 4: t-SNE Visualization of multimodal outlier synthesis results on the HMDB51 dataset. Our Feature Mixing excels at generating outlier samples by spanning wider embedding spaces without injecting noise at an extremely fast speed.

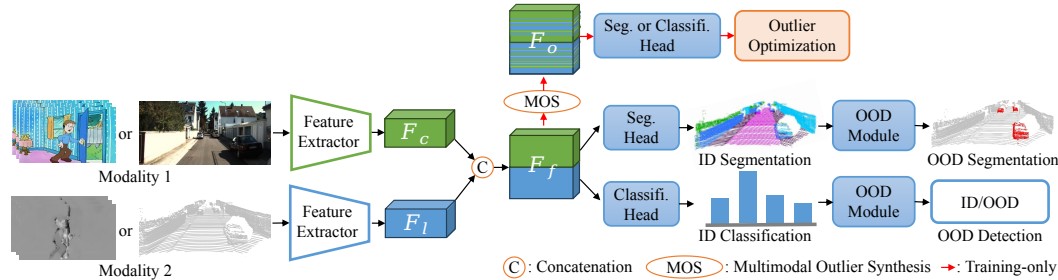

Figure 5: Overview of the proposed framework that integrates Feature Mixing for multimodal OOD detection and segmentation.

multimodal outlier features $\mathbf{F}_o = [\widetilde{\mathbf{F}}_c; \widetilde{\mathbf{F}}_l]$. Fig. 3 and Algorithm 1 provide detailed illustrations of the outlier synthesis process.

As shown in Fig. 4 (d), Feature Mixing excels at generating multimodal outliers by covering a broader embedding space without introducing noisy samples. The generated outliers exhibit *two key properties*: (1) These outliers share the same embedding space with the ID features but lie in low-likelihood regions (Theorem 1). (2) Their deviation from ID features is bounded, preventing excessive shifts while maintaining diversity (Theorem 2). These properties ensure that the outliers align with real OOD characteristics and can be supported by the following theorems. Due to space limits, the proofs are provided in the Appendix.

---

**Algorithm 1** Feature Mixing

**Input:** ID feature $\mathbf{F} = [\mathbf{F}_c; \mathbf{F}_l]$, where $\mathbf{F}_c$ is from modality 1 with $N_c$ channels, $\mathbf{F}_l$ is from modality 2 with $N_l$ channels; number of selected feature dimensions for mixing $N$.

**Python-like Code:**
$$select_c = random.sample(range(N_c), N)$$
$$select_l = random.sample(range(N_l), N)$$
$$\widetilde{\mathbf{F}}_c = \mathbf{F}_c.clone()$$
$$\widetilde{\mathbf{F}}_l = \mathbf{F}_l.clone()$$
$$\widetilde{\mathbf{F}}_c[select_c, :, :] = \mathbf{F}_l[select_l, :, :]$$
$$\widetilde{\mathbf{F}}_l[select_l, :, :] = \mathbf{F}_c[select_c, :, :]$$
$$\mathbf{F}_o = torch.cat([\widetilde{\mathbf{F}}_c, \widetilde{\mathbf{F}}_l], dim = 0)$$
**Output:** Multimodal outlier feature $\mathbf{F}_o$.

---

**Theorem 1.** *Outliers $\mathbf{F}_o$ synthesized by Feature Mixing lie in low-likelihood regions of the distribution of the ID features $\mathbf{F}$, complying with the criterion for real outliers.*

**Theorem 2.** *Outliers $\mathbf{F}_o$ are bounded in their deviation from $\mathbf{F}$, such that $|\mathbf{F}_o - \mathbf{F}|_2 \leq \sqrt{2N} \cdot \delta$, where $\delta = \max_{i,j} \left| \mathbf{F}_c^{(i)} - \mathbf{F}_l^{(j)} \right|$.*

Feature Mixing enables the online generation of multimodal outlier features and can be seamlessly integrated into existing training pipelines (Sec. 3.4). Besides, its simplicity makes it *modality-agnostic*, allowing application to diverse multimodal setups, such as images and point clouds or video and optical flow.

### 3.4 Framework for Outlier Optimization

Multimodal outlier features generated by Feature Mixing can be optimized using entropy maximization to help the model better distinguish between ID and OOD data, similar to outlier exposure-based methods [25, 38, 31]. Fig. 5 illustrates our framework, which integrates Feature Mixing into multi-

modal OOD detection and segmentation, comprising two key components: *Basic Multimodal Fusion* and *Multimodal Outlier Synthesis and Optimization*.

**Basic Multimodal Fusion.** Our framework employs a dual-stream network to extract features from different modalities using separate backbones. For example, for multimodal OOD segmentation, features extracted from the image and point cloud backbones, denoted as $\mathbf{F}_c \in \mathbb{R}^{N_c \times H \times W}$ and $\mathbf{F}_l \in \mathbb{R}^{N_l \times H \times W}$ respectively, are concatenated to form the fused representation $F_f$. $F_f$ contains both 2D and 3D scene information, which is then passed to a segmentation head for ID class segmentation. To enable efficient OOD segmentation at inference, we append an uncertainty-based OOD detection module to compute confidence scores for each prediction. This module supports various post-hoc OOD scoring methods [27, 24, 36], offering flexibility in design choices. Furthermore, the simple late-fusion design facilitates the integration of advanced cross-modal training strategies [26, 17] and generalizes easily to other modalities and tasks.

**Multimodal Outlier Synthesis and Optimization.** To mitigate overconfidence in uncertainty-based OOD detection, we incorporate outlier samples during training. These can be generated using existing methods such as Mixup [52], VOS [19], NP-Mix [17], or our proposed Feature Mixing. We then apply entropy-based optimization in Eq. (5) to maximize the entropy of outlier features. In this way, we can better separate the confidence scores between ID and OOD samples (Fig. 2), thereby improving the model's ability to distinguish OOD samples.

### 3.5 Training Strategy

Our training objective is to enhance OOD detection and segmentation while maintaining strong ID classification and segmentation performance.

**Multimodal OOD Segmentation.** Since accurate ID segmentation is crucial for the effectiveness of post-hoc OOD detection methods, optimizing ID segmentation is a priority. We employ focal loss [35] and Lovász-softmax loss [4], which are widely used in existing segmentation work [10, 53]. The focal loss $\mathcal{L}_{foc}$ addresses class imbalance by focusing on hard examples and is defined as:

$$\mathcal{L}_{foc} = \frac{1}{M} \sum_{m=1}^{M} \sum_{c=1}^{C} \alpha_c \mathbb{1}\{y_m = c\} FL(\mathbf{O}_{m,c}), \tag{2}$$

where $FL(p) = -(1-p)^\lambda \log(p)$ denotes the focal loss function and $\alpha_c$ is the weight w.r.t the $c$-th class. $\mathbb{1}\{\cdot\}$ is the indicator function. The Lovász-Softmax loss $\mathcal{L}_{lov}$ directly optimizes the mean IoU and is expressed as:

$$\mathcal{L}_{lov} = \frac{1}{C} \sum_{c=1}^{C} \overline{\Delta_{J_c}}(\mathbf{m}(c)), \tag{3}$$

where

$$\mathbf{m}_m(c) = \begin{cases} 1 - \mathbf{O}_{m,c} & \text{if } c = y_m, \\ \mathbf{O}_{m,c} & \text{otherwise.} \end{cases} \tag{4}$$

$\overline{\Delta_{J_c}}$ indicates the Lovász extension of the Jaccard index for class $c$. $\mathbf{m}(c) \in [0,1]^M$ indicates the vector of errors. For the generated multimodal outlier feature $\mathbf{F}_o$, we obtain a prediction output $\widetilde{\mathbf{O}} \in \mathbb{R}^{M \times C}$ using the segmentation head and aim to maximize the prediction entropy of the outlier features:

$$\mathcal{L}_{ent} = \frac{1}{M} \sum_{m=1}^{M} \sum_{c=1}^{C} \widetilde{\mathbf{O}}_{m,c} \log \widetilde{\mathbf{O}}_{m,c}. \tag{5}$$

The final loss is defined as:

$$\mathcal{L} = \mathcal{L}_{foc} + \mathcal{L}_{lov} + \gamma_1 \mathcal{L}_{ent}, \tag{6}$$

where $\gamma_1$ is a weighting factor that balances the contributions of each loss term.

**Multimodal OOD Detection.** The OOD detection loss combines classification loss with entropy regularization, which is defined as:

$$\mathcal{L} = \mathcal{L}_{cls} + \gamma_1 \mathcal{L}_{ent}, \tag{7}$$

where the cross-entropy loss is used for $\mathcal{L}_{cls}$.

# 4 Experiments

We evaluate Feature Mixing across eight datasets and four modalities to demonstrate its versatility. Specifically, we use nuScenes, SemanticKITTI, and our CARLA-OOD dataset for **Multimodal OOD Segmentation** using image and point cloud data. Additionally, we utilize five action recognition datasets from MultiOOD benchmark [17] for **Multimodal OOD Detection**, employing video and optical flow modalities.

## 4.1 Experimental Setup

**Datasets and Settings.** For multimodal OOD segmentation, we follow [17] to treat all vehicle classes as OOD on the SemanticKITTI [3] and nuScenes [6] datasets. During training, the labels of OOD classes are set to void and ignored. During inference, we aim to segment ID classes with high Intersection over Union (IoU) while detecting OOD classes as *unknown*.

We also introduce the **CARLA-OOD** dataset, created using the CARLA simulator [18], which includes RGB images, LiDAR point clouds, and 3D semantic segmentation ground truth, comprising a total of 245 samples. We select 34 anomalous objects as OOD, which are randomly positioned in front of the ego-vehicle across varied scenes and weather conditions, as shown in Fig. 6. Further details on CARLA-OOD are provided in the Appendix. The model is trained on the KITTI-CARLA [12] dataset with the same sensor setup and evaluated on CARLA-OOD with OOD objects. For multimodal OOD detection, we use HMDB51 [30], UCF101 [45], Kinetics-600 [28], HAC [16], and EPIC-Kitchens [11] datasets from the MultiOOD [17] benchmark. We evaluate using video and optical flow, where we train the model on one ID dataset and treat other datasets as OOD during testing.

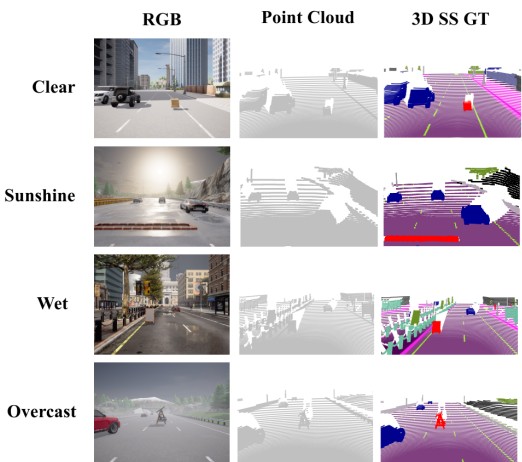

Figure 6: The proposed CARLA-OOD dataset for multimodal OOD segmentation. Points with red color are OOD objects.

**Implementation Details.** For multimodal OOD segmentation, our implementation follows [17] to build upon the fusion framework proposed in PMF [53], utilizing ResNet-34 [22] as the camera backbone and SalsaNext [10] as the LiDAR backbone. After feature extraction and fusion, we employ two 2D convolution layers as the segmentation head for ID segmentation. The OOD detection module uses MaxLogit [23] as the default scoring function. For multimodal OOD detection, we adopt the framework proposed in MultiOOD [17] and replace the multimodal outlier generation method with our Feature Mixing. Additional implementation details are provided in the Appendix.

**Evaluation Metrics.** For OOD segmentation, we evaluate both closed-set and OOD segmentation performance at the point level. For closed-set evaluation, we use the mean Intersection over Union for known classes ($mIoU_c$). For OOD performance, we report the area under the receiver operating characteristic curve (AUROC), the area under the precision-recall curve (AUPR), and the false positive rate at 95% true positive rate (FPR@95). For multimodal OOD detection, we report average accuracy (ACC) instead of $mIoU_c$ for closed-set evaluation, as well as AUROC and FPR@95 for OOD performance.

## 4.2 Main Results

### 4.2.1 Evaluation on Multimodal OOD Segmentation

We first evaluate our method on multimodal OOD segmentation. For baselines without outlier optimization, we consider basic Late Fusion, A2D [17] and xMUDA [26], with A2D being the state-of-the-art method. For baselines incorporating outlier optimization, we integrate Mixup [52], NP-Mix [17], and our Feature Mixing into the framework. Due to its inefficiency, VOS [19] is

| Method | SemanticKITTI | | | | nuScenes | | | | CARLA-OOD | | | |
|---|---|---|---|---|---|---|---|---|---|---|---|---|
| | FPR@95↓ | AUROC↑ | AUPR↑ | mIoU$_c$↑ | FPR@95↓ | AUROC↑ | AUPR↑ | mIoU$_c$↑ | FPR@95↓ | AUROC↑ | AUPR↑ | mIoU$_c$↑ |
| *w/o Outlier Optimization* | | | | | | | | | | | | |
| Late Fusion | 53.43 | 86.98 | 46.02 | 61.43 | 47.55 | 82.60 | 26.42 | 76.79 | 98.83 | 57.24 | 20.56 | 61.84 |
| xMUDA [26] | 55.37 | 89.67 | 51.41 | 60.61 | 44.32 | 83.47 | 20.20 | 78.79 | 97.00 | 57.86 | 10.35 | 65.15 |
| A2D [17] | 49.02 | 91.12 | 55.44 | 61.98 | 44.27 | 83.43 | 23.55 | 77.69 | 97.98 | 64.21 | 22.45 | 63.79 |
| *w/ Outlier Optimization* | | | | | | | | | | | | |
| Mixup [52] | 52.04 | 86.81 | 48.05 | 61.36 | 42.94 | 83.82 | 27.89 | 75.67 | 99.23 | 57.94 | 9.02 | 62.07 |
| NP-Mix [17] | 48.57 | 90.93 | 56.85 | 60.37 | 41.69 | 84.88 | 28.54 | 76.16 | 41.81 | 88.45 | 29.68 | 62.56 |
| Feature Mixing (ours) | 38.10 | 91.47 | 58.74 | 61.18 | 40.48 | 86.83 | **38.80** | 77.61 | **25.85** | 92.98 | 33.37 | 63.38 |
| xMUDA + FM (ours) | 36.63 | 91.54 | 53.89 | 60.43 | 39.49 | 85.29 | 28.74 | 77.69 | 30.35 | 92.45 | 33.44 | 65.92 |
| A2D + FM (ours) | **31.76** | **92.83** | **61.99** | 60.41 | **32.92** | **87.55** | 29.39 | 76.47 | 25.95 | **93.37** | **37.28** | 66.41 |

Table 1: Evaluation results on multimodal OOD segmentation datasets. FM: Feature Mixing.

| | OOD Datasets | | | | | | | | | | |
|---|---|---|---|---|---|---|---|---|---|---|---|
| Methods | Kinetics-600 | | UCF101 | | EPIC-Kitchens | | HAC | | Average | | ID ACC ↑ |
| | FPR@95↓ | AUROC↑ | FPR@95↓ | AUROC↑ | FPR@95↓ | AUROC↑ | FPR@95↓ | AUROC↑ | FPR@95↓ | AUROC↑ | |
| Baseline | 32.95 | 92.48 | 44.93 | 87.95 | 8.10 | 97.70 | 32.95 | 92.28 | 29.73 | 92.60 | 87.23 |
| Mixup [52] | 25.31 | 94.10 | 36.37 | 90.49 | 14.37 | 96.40 | 22.57 | 94.85 | 24.67 | 93.96 | 86.89 |
| VOS [19] | 31.70 | 93.22 | 38.77 | 89.93 | 15.39 | 96.82 | 31.58 | 93.03 | 29.36 | 93.25 | 87.34 |
| NPOS [48] | 25.31 | 93.94 | 37.17 | 89.71 | 13.00 | 96.50 | 24.17 | 93.94 | 24.91 | 93.52 | 87.12 |
| NP-Mix [17] | 24.52 | 93.96 | 36.49 | 89.67 | 6.96 | 97.53 | 22.92 | 94.41 | 22.72 | 93.89 | 86.89 |
| Feature Mixing (ours) | 19.61 | 94.72 | 34.32 | 90.06 | 10.15 | 96.34 | 15.96 | 95.54 | **20.01** | **94.17** | 87.00 |

Table 2: Multimodal OOD Detection using video and optical flow, with **HMDB51** as ID. Energy is used as the OOD score.

excluded from the segmentation task. Additionally, we combine A2D and xMUDA with Feature Mixing to demonstrate its versatility.

As shown in Tab. 1, Late Fusion without outlier optimization suffers from overconfidence, leading to high FPR@95 values, indicating poor ID-OOD separation. While A2D improves performance in most cases, it remains suboptimal. Integrating outlier optimization yields significant improvements for both NP-Mix and Feature Mixing, underscoring the importance of outlier synthesis. On SemanticKITTI, Feature Mixing improves Late Fusion by $15.33\%$ on FPR@95, $4.49\%$ on AUROC, and $12.72\%$ on AUPR. On nuScenes, Feature Mixing improves the Late Fusion baseline by $7.07\%$ on FPR@95, $4.23\%$ on AUROC, and $12.38\%$ on AUPR. At the same time, Feature Mixing introduces a negligible negative impact on mIoU$_c$ value.

Notably, all baselines without outlier optimization perform poorly on CARLA-OOD, with FPR@95 exceeding $97\%$, highlighting the dataset's difficulty and the overconfidence issue in uncertainty-based OOD methods. Feature Mixing significantly enhances Late Fusion on CARLA-OOD, reducing FPR@95 by $72.98\%$, improving AUROC by $35.74\%$, and increasing AUPR by $12.81\%$. Furthermore, A2D + Feature Mixing achieves the best results in most cases, demonstrating our framework's adaptability to advanced cross-modal training strategies.

### 4.2.2 Evaluation on Multimodal OOD Detection

To assess the generalizability of Feature Mixing across tasks and modalities, we evaluate it on MultiOOD for multimodal OOD detection in action recognition, where video and optical flow serve as distinct modalities. We replace the outlier generation method in MultiOOD framework with Feature Mixing and compare it against Mixup [52], VOS [19], NPOS [48], and NP-Mix [17]. Models are trained on HMDB51 [30] or Kinetics-600 [39], and other datasets are treated as OOD during testing. As shown in Tab. 2, our Feature Mixing outperforms other outlier generation methods in most cases, achieving the lowest FPR@95 of $20.01\%$ and the highest AUROC of $94.17\%$ on average when using HMDB51 as ID. Due to space limits, we put the results on Kinetics-600 in the Appendix. These results highlight the effectiveness of Feature Mixing in improving OOD detection across various tasks and modalities. Similarly, Feature Mixing introduces a negligible impact on ID ACC.

### 4.3 Ablation Studies

**Computational Cost.** Tab. 3 compares the computational cost of different outlier synthesis methods. For OOD detection, the reported time corresponds to generating 2048 multimodal outlier samples of shape 4352. For OOD segmentation, it represents the time to synthesize $256 \times 352$ samples of shape 48.

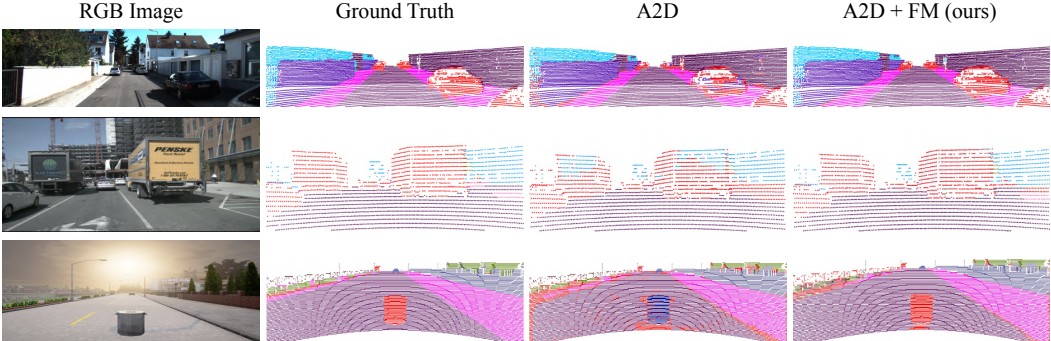

| RGB Image | Ground Truth | A2D | A2D + FM (ours) |

Figure 7: Visualization results on different datasets. From top to down: SemanticKITTI, nuScenes, and CARLA-OOD. Points with red color are OOD objects. Our method can segment OOD objects accurately. More visualization results are in the Appendix.

While Mixup [52] is efficient, it performs poorly in OOD detection. In contrast, NP-Mix [17] achieves strong performance but is computationally expensive. Feature Mixing, benefiting from its simple design, is both highly efficient and effective. Compared to NP-Mix, Feature Mixing provides a $10\times$ speedup for multimodal OOD detection and a $370\times$ speedup for segmentation, making it well-suited for real-world applications.

|  | OOD Detection | OOD Segmentation |
|---|---|---|
| Mixup [52] | 0.038 | 0.019 |
| VOS [19] | 152.05 | 61.49 |
| NP-Mix [17] | 0.545 | 4.81 |
| Feature Mixing (ours) | 0.058 | 0.013 |

Table 3: Computational cost of outlier synthesis methods (s).

**Visualization.** Fig. 7 presents the visualization of OOD segmentation results across different datasets, showcasing the RGB image, 3D semantic ground truth, and predictions from A2D and our best-performing A2D+FM model. The baseline method A2D struggles to identify OOD objects, whereas our method accurately segments OOD with minimal noise, demonstrating the effectiveness of the proposed framework. Additional visualizations can be found in the Appendix.

**Hyperparameter Sensitivity.** We evaluate the sensitivity of Feature Mixing to the hyperparameter $N$, using HMDB51 as ID dataset and Kinetics-600 as OOD dataset. Our findings, as illustrated in Fig. 8, demonstrate that Feature Mixing is robust and consistently outperforms the baseline across all parameter settings.

**Feature Mixing in Tri- and Unimodal Settings.** To further demonstrate the generality of our method, we conduct additional experiments using Feature Mixing (FM) in both tri-modal and unimodal settings.

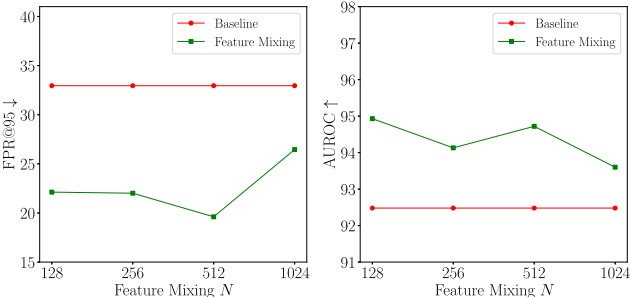

Figure 8: Ablation on the number $N$ in Feature Mixing.

*Tri-Modal Setting (Video + Optical Flow + Audio).* We extend FM to a tri-modal setting on the EPIC-Kitchens dataset, where modalities include video, optical flow, and audio. FM is applied by randomly selecting a subset of $N$ feature dimensions from each modality and performing a cyclic swap: video $\rightarrow$ audio, audio $\rightarrow$ optical flow, and optical flow $\rightarrow$ video. This procedure synthesizes outlier features for all three modalities. As shown in Tab. 4a, FM consistently outperforms NP-Mix across all evaluation metrics, demonstrating robustness and versatility in complex multimodal scenarios.

*Unimodal Setting (Video Only).* FM is inherently modality-agnostic, operating purely in feature space without relying on input-specific assumptions. This makes it naturally extendable to unimodal scenarios. We implement a unimodal variant by splitting the video feature embedding into two halves and randomly swapping $N$ dimensions between them, effectively synthesizing outliers within a single

| Method | FPR@95↓ | AUROC↑ | ACC↑ |
|---|---|---|---|
| Baseline | 69.22 | 72.39 | 73.13 |
| NP-Mix | 62.69 | 74.95 | 71.46 |
| Feature Mixing | **61.38** | **77.23** | **73.69** |

(a) EPIC-Kitchens (Tri-modal)

| Method | FPR@95↓ | AUROC↑ | ACC↑ |
|---|---|---|---|
| Baseline | 64.05 | 83.14 | 86.93 |
| NP-Mix | 62.53 | 83.62 | 87.15 |
| Feature Mixing | **57.30** | **84.41** | **88.89** |

(b) HMDB51 (Unimodal)

Table 4: OOD detection results with Feature Mixing in tri-modal and unimodal settings.

modality. Evaluated on the HMDB51 dataset, this approach outperforms the NP-Mix baseline across all metrics, confirming FM's applicability even in the absence of multiple modalities (Tab. 4b).

**Different Classes as OOD.** For multimodal OOD segmentation, we follow [17] and designate all vehicle classes as OOD. Here, we experiment with different OOD category assignments: "ground" (road, sidewalk, parking, otherground), "structure" (building, other-structure), and "nature" (vegetation, trunk, terrain). As shown in Tab. 5, Feature Mixing remains robust across all splits, consistently outperforming the baseline by a significant margin.

| Method | FPR@95↓ | AUROC↑ | AUPR↑ | mIoU$_c$↑ |
|---|---|---|---|---|
| *"ground" classes as OOD* | | | | |
| A2D [17] | 71.15 | 74.92 | 69.13 | 66.57 |
| A2D + NP-Mix | 53.60 | 94.71 | 95.30 | 65.07 |
| A2D + FM (ours) | 36.30 | 95.89 | 96.04 | 65.88 |
| *"structure" classes as OOD* | | | | |
| A2D [17] | 23.50 | 95.20 | 75.23 | 61.38 |
| A2D + NP-Mix | 22.14 | 95.41 | 76.36 | 60.54 |
| A2D + FM (ours) | 18.05 | 96.09 | 79.85 | 61.79 |
| *"nature" classes as OOD* | | | | |
| A2D [17] | 37.97 | 92.74 | 90.22 | 62.76 |
| A2D + NP-Mix | 30.88 | 95.18 | 92.51 | 61.21 |
| A2D + FM (ours) | 20.60 | 96.13 | 94.08 | 62.67 |

Table 5: Ablation on different classes as OOD on SemanticKITTI.

## 5   Conclusion

In this work, we introduce Feature Mixing, an extremely simple and fast method for multimodal outlier synthesis with theoretical support. Feature Mixing is modality-agnostic and applicable to various modality combinations. Moreover, its lightweight design achieves a $10\times$ to $370\times$ speedup over existing methods while maintaining strong OOD performance. To mitigate overconfidence, outlier features are optimized via entropy maximization within our framework. Additionally, we present CARLA-OOD, a challenging multimodal dataset featuring synthetic OOD objects captured under diverse scenes and weather conditions. Extensive experiments across eight datasets and four modalities validate the versatility and effectiveness of Feature Mixing and our proposed framework.

**Limitations and future work.** Feature Mixing uses a random selection of feature dimensions to swap between modalities. While this is highly efficient and theoretically grounded, it may not always target the most informative feature regions for outlier synthesis. Future work could explore *adaptive or learnable selection mechanisms* that dynamically identify features that maximize OOD separability.

**Societal impact.** In our work, Feature Mixing advances multimodal OOD detection and segmentation by enabling more effective and efficient outlier exposure during training. This contributes directly to the reliability and safety of autonomous systems, including self-driving cars, by helping models handle unfamiliar or unexpected scenarios in open-world environments. Beyond autonomous driving, the method also has broader societal impacts in other safety-critical domains such as healthcare and security, where robustness to unknown or anomalous data is crucial. By improving model reliability in the presence of distribution shifts, our approach supports the deployment of AI systems in dynamic, high-stakes environments where failure could have significant consequences.

## Acknowledgments

The authors acknowledge that this work was supported by the Federal Ministry of Education and Research (BMBF) as part of MANNHEIM-AutoDevSafeOps (reference number 01IS22087E). It was also funded by the Bavarian Ministry for Economic Affairs, Regional Development and Energy as part of a project to support the thematic development of the Institute for Cognitive Systems. The authors also acknowledge the support of "In-service diagnostics of the catenary/pantograph and wheelset axle systems through intelligent algorithms" (SENTINEL) project, supported by the ETH Mobility Initiative.

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

Figure 9: Feature Mixing introduces bias that shifts $\mathbf{F}_o$ away from the ID distribution $\mathbf{F}$, which is proportional to $N$. A large $N$ promotes outliers to lie in low-likelihood regions of the ID feature distribution. Therefore, $N$ can be adjusted to generate targeted outliers.

## A Theoretical Insights for Feature Mixing

**Problem Setup.** Let $\mathbf{F} = \begin{bmatrix} \mathbf{F}_c \\ \mathbf{F}_l \end{bmatrix} \in \mathbb{R}^{2d}$ be the concatenated in-distribution (ID) features from two modalities, where $\mathbf{F}_c \sim P$ with mean $\boldsymbol{\mu}_c$ and covariance $\boldsymbol{\Sigma}_c$, $\mathbf{F}_l \sim Q$ with mean $\boldsymbol{\mu}_l$ and covariance $\boldsymbol{\Sigma}_l$, $\boldsymbol{\mu}_c \neq \boldsymbol{\mu}_l$. The joint distribution of $\mathbf{F}$ has mean $\boldsymbol{\mu} = \begin{bmatrix} \boldsymbol{\mu}_c \\ \boldsymbol{\mu}_l \end{bmatrix}$ and covariance $\boldsymbol{\Sigma} = \begin{bmatrix} \boldsymbol{\Sigma}_c & \boldsymbol{\Sigma}_{cl} \\ \boldsymbol{\Sigma}_{cl}^T & \boldsymbol{\Sigma}_l \end{bmatrix}$, where $\boldsymbol{\Sigma}_{cl}$ encodes cross-modal dependencies.

**Feature Mixing** swaps $N$ features between $\mathbf{F}_c$ and $\mathbf{F}_l$ to generate perturbed features $\widetilde{\mathbf{F}}_c$ and $\widetilde{\mathbf{F}}_l$, then concatenates them to form $\mathbf{F}_o = \begin{bmatrix} \widetilde{\mathbf{F}}_c \\ \widetilde{\mathbf{F}}_l \end{bmatrix}$, which can be written as:

$$\widetilde{\mathbf{F}}_c = \mathbf{F}_c \odot (1 - \mathbf{M_1}) + \mathbf{F}_l \odot \mathbf{M_1}, \tag{8}$$

$$\widetilde{\mathbf{F}}_l = \mathbf{F}_l \odot (1 - \mathbf{M_2}) + \mathbf{F}_c \odot \mathbf{M_2}, \tag{9}$$

where $\mathbf{M_1}, \mathbf{M_2} \in \{0, 1\}^d$ is a binary mask with $N$ ones.

**Theorem 1** *Outliers $\mathbf{F}_o$ synthesized by Feature Mixing lie in low-likelihood regions of the distribution of the ID features $\mathbf{F}$, complying with the criterion for real outliers.*

**Proof 1:** After swapping $N$ features, the feature of each modality has a $\frac{N}{d}$ probability of being swapped from the other modality. Therefore, the mean of $\widetilde{\mathbf{F}}_c$ is a weighted average of $\boldsymbol{\mu}_c$ and $\boldsymbol{\mu}_l$, and similarly for $\widetilde{\mathbf{F}}_l$:

$$\mathbb{E}[\widetilde{\mathbf{F}}_c] = \left(1 - \frac{N}{d}\right)\boldsymbol{\mu}_c + \frac{N}{d}\boldsymbol{\mu}_l, \tag{10}$$

$$\mathbb{E}[\widetilde{\mathbf{F}}_l] = \left(1 - \frac{N}{d}\right)\boldsymbol{\mu}_l + \frac{N}{d}\boldsymbol{\mu}_c. \tag{11}$$

The perturbed mean $\boldsymbol{\mu}_o$ of $\mathbf{F}_o$ becomes a weighted combination of the original modality means:

$$\boldsymbol{\mu}_o = \mathbb{E}[\mathbf{F}_o] = \begin{bmatrix} \mathbb{E}[\widetilde{\mathbf{F}}_c] \\ \mathbb{E}[\widetilde{\mathbf{F}}_l] \end{bmatrix} = \begin{bmatrix} \left(1 - \frac{N}{d}\right)\boldsymbol{\mu}_c + \frac{N}{d}\boldsymbol{\mu}_l \\ \left(1 - \frac{N}{d}\right)\boldsymbol{\mu}_l + \frac{N}{d}\boldsymbol{\mu}_c \end{bmatrix}. \tag{12}$$

The deviation from the original mean $\boldsymbol{\mu}$ becomes:

$$\Delta\boldsymbol{\mu} = \boldsymbol{\mu}_o - \boldsymbol{\mu} = \frac{N}{d}\begin{bmatrix} \boldsymbol{\mu}_l - \boldsymbol{\mu}_c \\ \boldsymbol{\mu}_c - \boldsymbol{\mu}_l \end{bmatrix}. \tag{13}$$

Since $\boldsymbol{\mu}_c \neq \boldsymbol{\mu}_l$, $\Delta\boldsymbol{\mu} \neq 0$, introducing a bias that shifts $\mathbf{F}_o$ away from the ID distribution proportional to $N$ (Fig. 9) and $|\boldsymbol{\mu}_c - \boldsymbol{\mu}_l|$. The Mahalanobis distance measures how far $\mathbf{F}_o$ deviates from the mean $\boldsymbol{\mu}$ of the original joint distribution $\mathbf{F}$, weighted by the inverse covariance $\boldsymbol{\Sigma}^{-1}$:

$$D^2(\mathbf{F}_o) = (\mathbf{F}_o - \boldsymbol{\mu})^T\boldsymbol{\Sigma}^{-1}(\mathbf{F}_o - \boldsymbol{\mu}). \tag{14}$$

By defining $\mathbf{F}_o - \boldsymbol{\mu}$ as $(\mathbf{F}_o - \boldsymbol{\mu}_o) + (\boldsymbol{\mu}_o - \boldsymbol{\mu})$, we get:

$$D^2(\mathbf{F}_o) = (\mathbf{F}_o - \boldsymbol{\mu}_o + \Delta\boldsymbol{\mu})^T\boldsymbol{\Sigma}^{-1}(\mathbf{F}_o - \boldsymbol{\mu}_o + \Delta\boldsymbol{\mu}). \tag{15}$$

After expanding the quadratic form, we get:

$$D^2(\mathbf{F}_o) = (\mathbf{F}_o - \boldsymbol{\mu}_o)^T\boldsymbol{\Sigma}^{-1}(\mathbf{F}_o - \boldsymbol{\mu}_o) + (\Delta\boldsymbol{\mu})^T\boldsymbol{\Sigma}^{-1}\Delta\boldsymbol{\mu} + \\ 2(\Delta\boldsymbol{\mu})^T\boldsymbol{\Sigma}^{-1}(\mathbf{F}_o - \boldsymbol{\mu}_o). \tag{16}$$

The first term captures the deviation of $\mathbf{F}_o$ from its perturbed mean $\boldsymbol{\mu}_o$, weighted by $\boldsymbol{\Sigma}^{-1}$. The second term is the bias from the mean shift $\Delta\boldsymbol{\mu}$, which grows with $N$ and $|\boldsymbol{\mu}_c - \boldsymbol{\mu}_l|$. The last term is the cross-term that involves both perturbation noise and mean shift.

The original covariance $\boldsymbol{\Sigma}$ encodes intra- and cross-modal correlations. After swapping, $\text{Cov}(\widetilde{\mathbf{F}}_c)$ becomes a mix of $\boldsymbol{\Sigma}_c$ and $\boldsymbol{\Sigma}_l$, similarly for $\text{Cov}(\widetilde{\mathbf{F}}_l)$. Besides, swapped features disrupt dependencies between $\mathbf{F}_c$ and $\mathbf{F}_l$, invalidating $\boldsymbol{\Sigma}_{cl}$. Therefore, the perturbed features $\mathbf{F}_o$ have a new covariance structure $\boldsymbol{\Sigma}_o \neq \boldsymbol{\Sigma}$ and this mismatch inflates the first term in Eq. (16). $\mathbf{F}_o - \boldsymbol{\mu}_o$ represents deviations under the perturbed distribution $\boldsymbol{\Sigma}_o$, which are not aligned with the original covariance structure $\boldsymbol{\Sigma}$. This misalignment causes $\boldsymbol{\Sigma}^{-1}$ to assign incorrect weights to the deviations, leading to larger values in the quadratic form. Besides, the mean shift $\Delta\boldsymbol{\mu}$ in the second term and the last cross-term can also lead to large values for $D^2(\mathbf{F}_o)$. For Gaussian-distributed $\mathbf{F}$, the likelihood of $\mathbf{F}_o$ decays exponentially with $D^2(\mathbf{F}_o)$:

$$p(\mathbf{F}_o) \propto \exp\left(-\frac{1}{2}D^2(\mathbf{F}_o)\right). \tag{17}$$

Therefore, the inflated $D^2(\mathbf{F}_o)$ from covariance mismatch, mean shift, and cross-term forces $p(\mathbf{F}_o)$ to be small, satisfying the low-likelihood criterion for outliers.

Here, we show mathematically that the expected Mahalanobis distance of $\mathbf{F}_o$ exceeds that of ID samples $\mathbf{F}$:

$$\mathbb{E}[D^2(\mathbf{F}_o)] > \mathbb{E}[D^2(\mathbf{F})], \tag{18}$$

where $D^2(\mathbf{x}) = (\mathbf{x} - \boldsymbol{\mu})^T\boldsymbol{\Sigma}^{-1}(\mathbf{x} - \boldsymbol{\mu})$. For ID samples $\mathbf{F}$, the squared Mahalanobis distance follows a chi-squared distribution with $2d$ degrees of freedom with expectation:

$$\mathbb{E}[D^2(\mathbf{F})] = \mathbb{E}\left[\text{Tr}\left(\boldsymbol{\Sigma}^{-1}(\mathbf{F} - \boldsymbol{\mu})(\mathbf{F} - \boldsymbol{\mu})^T\right)\right] \\ = \text{Tr}(\boldsymbol{\Sigma}^{-1}\boldsymbol{\Sigma}) = \text{Tr}(I_{2d}) = 2d, \tag{19}$$

where $\text{Tr}(\cdot)$ is the trace of a matrix. For outliers $\mathbf{F}_o$, from Eq. (16) we know $\mathbb{E}[D^2(\mathbf{F}_o)]$ is the sum of the expectation of three terms. Since $\mathbb{E}[\mathbf{F}_o - \boldsymbol{\mu}_o] = 0$, the expectation of the last term is $0$ and:

$$\mathbb{E}[D^2(\mathbf{F}_o)] = \text{Tr}(\boldsymbol{\Sigma}^{-1}\boldsymbol{\Sigma}_o) + (\boldsymbol{\mu}_o - \boldsymbol{\mu})^T\boldsymbol{\Sigma}^{-1}(\boldsymbol{\mu}_o - \boldsymbol{\mu}). \tag{20}$$

Let $\Delta\boldsymbol{\Sigma} = \boldsymbol{\Sigma}_o - \boldsymbol{\Sigma}$, the trace term becomes:

$$\text{Tr}(\boldsymbol{\Sigma}^{-1}\boldsymbol{\Sigma}_o) = \text{Tr}(\boldsymbol{\Sigma}^{-1}(\boldsymbol{\Sigma} + \Delta\boldsymbol{\Sigma})) = 2d + \text{Tr}(\boldsymbol{\Sigma}^{-1}\Delta\boldsymbol{\Sigma}). \tag{21}$$

Now we prove that Feature Mixing ensures $\text{Tr}(\boldsymbol{\Sigma}^{-1}\Delta\boldsymbol{\Sigma}) \geq 0$. Assume $\boldsymbol{\Sigma}$ is diagonal (without loss of generality via eigendecomposition) with $\boldsymbol{\Sigma} = \text{diag}(\sigma_1^2, \ldots, \sigma_{2d}^2)$. Feature Mixing increases variances in dimensions with low $\sigma_i^2$ and decreases them in dimensions with high $\sigma_i^2$. Let:

$$\Delta\boldsymbol{\Sigma} = \text{diag}(\Delta_1, \ldots, \Delta_{2d}), \quad \Delta_i = \sigma_{o,i}^2 - \sigma_i^2. \tag{22}$$

- For $\sigma_i^2 \le \sigma_{o,i}^2$: $\Delta_i \ge 0$, and $\frac{\Delta_i}{\sigma_i^2}$ is large.

- For $\sigma_i^2 > \sigma_{o,i}^2$: $\Delta_i \le 0$, and $\frac{\Delta_i}{\sigma_i^2}$ is small in magnitude.

The trace becomes:

$$\mathrm{Tr}(\mathbf{\Sigma}^{-1}\Delta\mathbf{\Sigma}) = \sum_{i=1}^{2d} \frac{\Delta_i}{\sigma_i^2}. \tag{23}$$

The positive terms dominate because $\mathbf{\Sigma}^{-1}$ weights low-variance dimensions more heavily. Thus, $\mathrm{Tr}(\mathbf{\Sigma}^{-1}\Delta\mathbf{\Sigma}) \ge 0$. For example, if $\sigma_i^2 = a$ increases to $\sigma_{o,i}^2 = b$ and $\sigma_j^2 = b$ decreases to $\sigma_{o,j}^2 = a$, where $0 < a \le b$:

$$\frac{\Delta_i}{\sigma_i^2} = \frac{b-a}{a}, \quad \frac{\Delta_j}{\sigma_j^2} = \frac{a-b}{b}$$
$$\Rightarrow \frac{\Delta_i}{\sigma_i^2} + \frac{\Delta_j}{\sigma_j^2} = \frac{(b-a)^2}{ab} \ge 0. \tag{24}$$

Since Feature Mixing randomly swaps features from two modalities, the probability of increasing or decreasing $\sigma_i^2$ is the same, and therefore $\mathrm{Tr}(\mathbf{\Sigma}^{-1}\Delta\mathbf{\Sigma}) \ge 0$. The second term in Eq. (20) is strictly positive for $\boldsymbol{\mu}_o \ne \boldsymbol{\mu}$:

$$(\boldsymbol{\mu}_o - \boldsymbol{\mu})^T \mathbf{\Sigma}^{-1} (\boldsymbol{\mu}_o - \boldsymbol{\mu}) > 0. \tag{25}$$

Combining all terms:

$$\mathbb{E}[D^2(\mathbf{F}_o)] = 2d + \underbrace{\mathrm{Tr}(\mathbf{\Sigma}^{-1}\Delta\mathbf{\Sigma})}_{\ge 0} + \underbrace{(\boldsymbol{\mu}_o - \boldsymbol{\mu})^T \mathbf{\Sigma}^{-1} (\boldsymbol{\mu}_o - \boldsymbol{\mu})}_{>0}$$
$$> 2d = \mathbb{E}[D^2(\mathbf{F})]. \tag{26}$$

Since for Gaussian-distributed $\mathbf{F}$, the likelihood of $\mathbf{F}_o$ decays exponentially with $D^2(\mathbf{F}_o)$, $p(\mathbf{F}_o)$ is much smaller than $p(\mathbf{F})$ and therefore $\mathbf{F}_o$ lie in low-likelihood regions.

**Theorem 2** *Outliers $\mathbf{F}_o$ are bounded in their deviation from $\mathbf{F}$, such that $|\mathbf{F}_o - \mathbf{F}|_2 \le \sqrt{2N} \cdot \delta$, where $\delta = \max_{i,j} \left| \mathbf{F}_c^{(i)} - \mathbf{F}_l^{(j)} \right|$.*

**Proof 2:** While $\mathbf{F}_o$ is statistically anomalous, it remains geometrically proximate to $\mathbf{F}$ and is bounded in their deviation from $\mathbf{F}$. The Euclidean distance between $\mathbf{F}_o$ and $\mathbf{F}$ is:

$$|\mathbf{F}_o - \mathbf{F}|_2 = \sqrt{|\widetilde{\mathbf{F}}_c - \mathbf{F}_c|_2^2 + |\widetilde{\mathbf{F}}_l - \mathbf{F}_l|_2^2}. \tag{27}$$

For each modality, the deviation is bounded by the maximum feature difference $\delta = \max_{i,j} \left| \mathbf{F}_c^{(i)} - \mathbf{F}_l^{(j)} \right|$:

$$|\widetilde{\mathbf{F}}_c - \mathbf{F}_c|_2 = |\mathbf{M} \odot (\mathbf{F}_l - \mathbf{F}_c)|_2 \le \sqrt{N} \cdot \delta, \tag{28}$$
$$|\widetilde{\mathbf{F}}_l - \mathbf{F}_l|_2 = |\mathbf{M} \odot (\mathbf{F}_c - \mathbf{F}_l)|_2 \le \sqrt{N} \cdot \delta. \tag{29}$$

Therefore:

$$|\mathbf{F}_o - \mathbf{F}|_2 \le \sqrt{(\sqrt{N} \cdot \delta)^2 + (\sqrt{N} \cdot \delta)^2} = \sqrt{2N} \cdot \delta. \tag{30}$$

Since $N \ll d$, $\sqrt{2N} \cdot \delta$ remains small, ensuring $\mathbf{F}_o$ stays near $\mathbf{F}$ and preserves *semantic consistency*. In conclusion, both $\mathbf{F}_o$ and $\mathbf{F}$ share the same embedding space, but $\mathbf{F}_o$ lies in low-likelihood regions of the distribution of $\mathbf{F}$.

# B   Additional Visualization Results

Fig. 10 to Fig. 12 present visualizations of multimodal OOD segmentation results across different datasets, showcasing the RGB image, 3D semantic ground truth, and predictions from various baselines. The baseline method struggles to identify OOD objects, whereas our method accurately segments OOD objects with minimal noise.

| RGB Image | Ground Truth | A2D | A2D + FM (ours) |

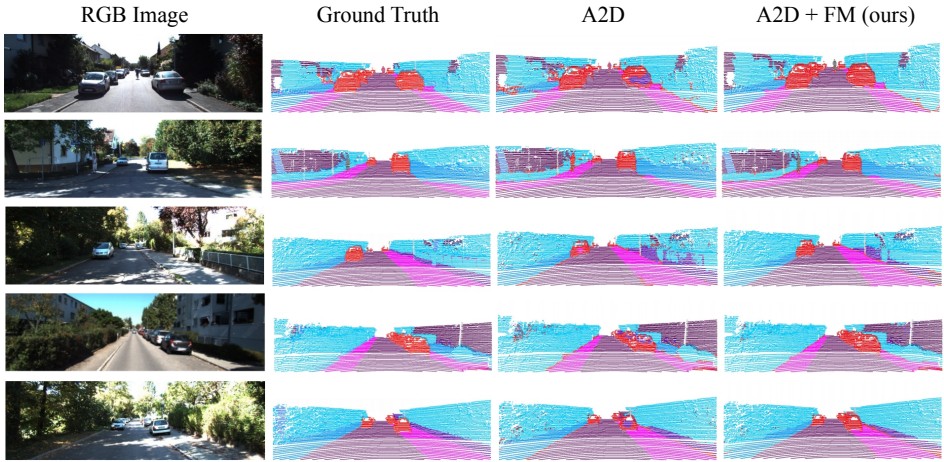

Figure 10: More visualizations on SemanticKITTI dataset. Points with red color are OOD objects. Our method accurately segments OOD objects, outperforming the baseline method.

| RGB Image | Ground Truth | A2D | A2D + FM (ours) |

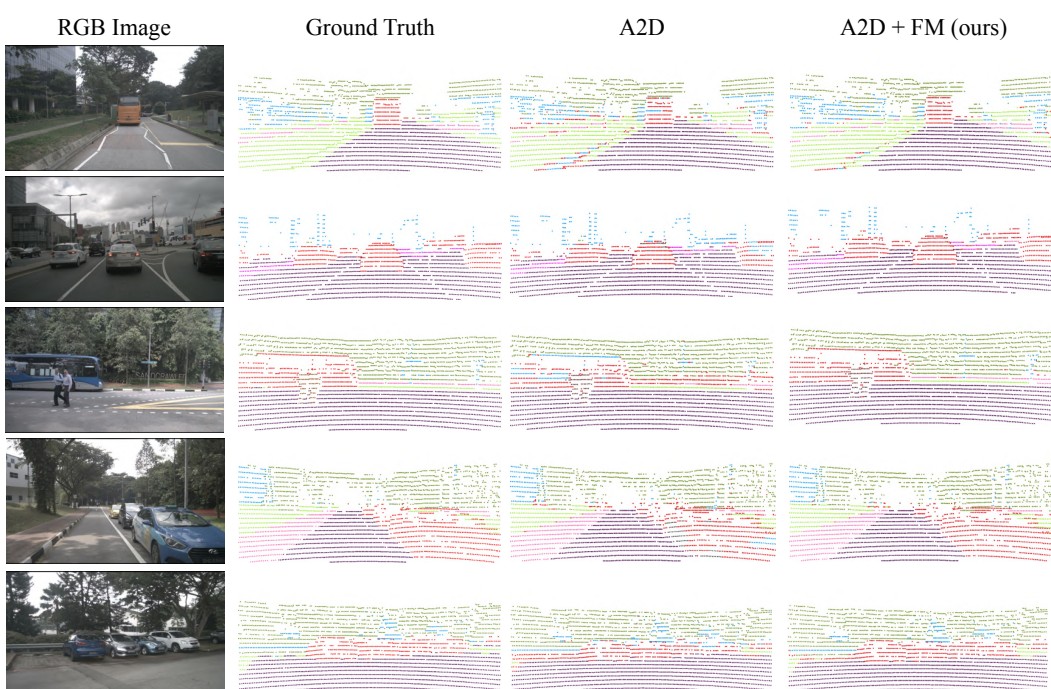

Figure 11: More visualizations on nuScenes dataset. Points with red color are OOD objects. Our method accurately segments OOD objects, outperforming the baseline method.

## C   More Details on Datasets

### C.1   Realistic Datasets

We evaluate our approach on two widely-used autonomous driving datasets, including nuScenes [6] and SemanticKITTI [3]. Both datasets provide paired LiDAR point cloud and RGB image, along with point-level semantic annotations. The nuScenes dataset contains $28,130$ training frames and $6,019$ validation frames, annotated with 16 semantic classes. SemanticKITTI consists of $21,000$ frames from sequences 00-10 for training and validation, annotated with 19 semantic classes. Following [53, 7], we use sequence 08 for validation. The remaining sequences (00-07 and 09-10) are used for training. In our OOD segmentation setting, we follow [17] to map all *vehicle* categories to a single *unknown* class to represent out-of-distribution (OOD) objects in both datasets. Specifically, in

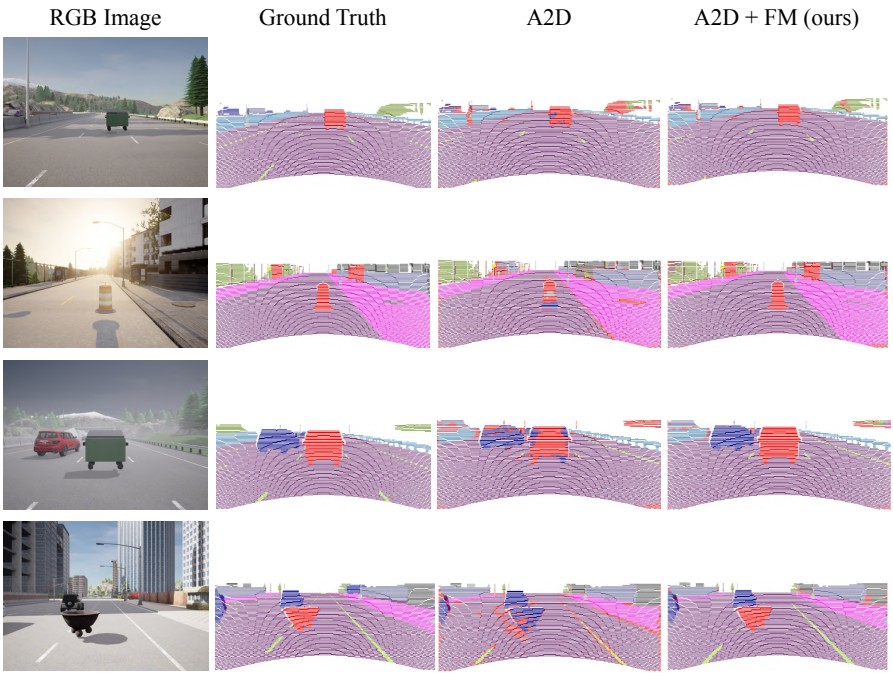

| RGB Image | Ground Truth | A2D | A2D + FM (ours) |

Figure 12: More visualizations on CARLA-OOD dataset. Points with red color are OOD objects. Our method accurately segments OOD objects, outperforming the baseline method.

SemanticKITTI, the categories *{car, bicycle, motorcycle, truck, and other-vehicle}* are remapped, while in nuScenes, the remapped categories include *{bicycle, bus, car, construction_vehicle, trailer, truck, and motorcycle}*. All other categories are retained as in-distribution (ID) classes and follow the standard segmentation settings. During training, we set the labels of OOD classes to void and ignore them. During inference, we aim to segment the ID classes with high Intersection over Union (IoU) and detect OOD classes as *unknown*.

## C.2 Synthetic Dataset

**Limitations of the realistic datasets.** The evaluation of multimodal OOD segmentation on datasets like nuScenes [6] and SemanticKITTI [3] often relies on manually remapping existing categories to simulate OOD classes. This methodology, while prevalent, suffers from two significant drawbacks. Firstly, the categories designated as OOD are often common objects (e.g., vehicles, ground, structures) that may not faithfully represent the characteristics of genuine, unseen anomalies encountered in real-world scenarios. Secondly, despite these designated OOD classes being nominally ignored during training (e.g., by excluding them from loss computation for known classes), the model is nevertheless exposed to these objects within the training data. This creates a substantial risk of data leakage, as the model may implicitly learn characteristics of these supposedly 'unseen' OOD classes.

Inspired by the development of existing 2D OOD segmentation benchmarks, where models are trained on Cityscapes [9] and tested on Fishyscapes [5] with the same class setup but additional synthetic OOD objects, we create the CARLA-OOD dataset for multimodal OOD segmentation task. We use the KITTI-CARLA dataset [12] to train the base model, which is generated using the CARLA simulator [18] with paired LiDAR and camera data. The CARLA-OOD dataset aligns with the KITTI-CARLA sensor configurations but incorporates randomly placed OOD objects in diverse scenes and weather conditions for testing. The KITTI-CARLA dataset consists of 7 sequences, each containing 5,000 frames captured from distinct CARLA maps and annotated with 22 classes for the LiDAR point cloud. We select 1,000 evenly sampled frames from each sequence, resulting in a total of 7,000 frames for training and validation. The dataset is split into a training set (Town01, Town03–Town07) and a validation set (Town02), with testing performed on our CARLA-OOD dataset. The CARLA-OOD dataset consists of 245 paired LiDAR and camera samples captured across 5 CARLA maps and 6 weather conditions, each sample containing at least one OOD object.

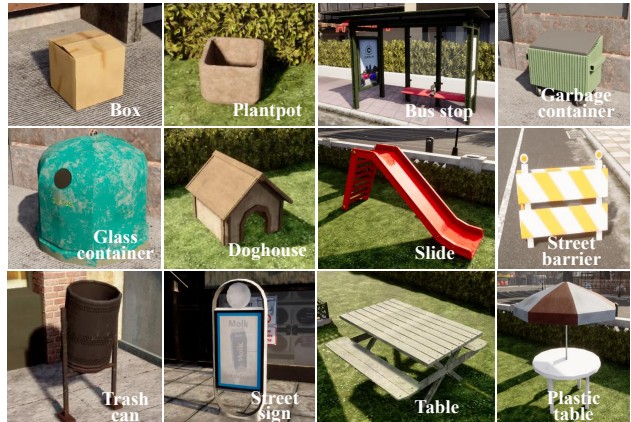

Figure 13: Example of OOD objects in our CARLA-OOD dataset.

| Sensor | Position (x, y, z) [meters] | Configurations |
|---|---|---|
| LiDAR | (0, 0, 1.80) | Channels: 64 |
| | | Range: 80.0 meters |
| | | Upper FOV: 2 degrees |
| | | Lower FOV: -24.8 degrees |
| RGB Camera | (0.30, 0, 1.70) | FOV: 72 degrees |
| Semantic Camera | (0.30, 0, 1.70) | FOV: 72 degrees |

Table 6: Sensor configurations for CARLA-OOD dataset.

To avoid class overlap with KITTI-CARLA, 34 OOD objects are selected and randomly placed within the scenes during dataset generation. The dataset is annotated with 22 classes aligned with KITTI-CARLA, along with an additional *unknown* class for OOD objects.

### C.3 Generation of CARLA-OOD Dataset

The CARLA-OOD dataset is created using the CARLA simulator, with a sensor setup aligned to the KITTI-CARLA dataset, consisting of a camera and a LiDAR on the ego-vehicle. Detailed sensor configurations are provided in Tab. 6, with positions defined relative to the ego-vehicle. Thirty-four obstacles from CARLA's dynamic and static classes are randomly placed in front of the ego-vehicle at varying distances as OOD objects (Fig. 13). The simulation spans diverse scenes (Town01, Town02, Town04, Town05, Town10) and weather conditions (e.g., clear, wet, foggy, sunshine, overcast), capturing both semantic and covariate shifts. The dataset includes RGB images with a resolution of $1392 * 1024$ pixels, LiDAR point cloud, point-level semantic labels, and transformation matrices between sensors.

### C.4 MultiOOD benchmark

MultiOOD [17] is the first benchmark designed for Multimodal OOD Detection, comprising five action recognition datasets (EPIC-Kitchens [11], HAC [16], HMDB51 [30], UCF101 [45], and Kinetics-600 [28]) with over $85,000$ video clips, where video, optical flow, and audio are used as different types of modalities. Fig. 14 shows an example of the Far-OOD setup in MultiOOD. This setup considers an entire dataset as in-distribution (ID) and further collects datasets, which comprise similar tasks but are disconnected from any ID categories, as OOD datasets. In this scenario, both semantic and domain shifts are present between the ID and OOD samples. We follow the

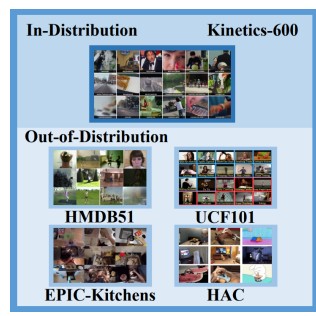

Figure 14: Multimodal Far-OOD setup in MultiOOD, where Kinetics-600 is the ID dataset and the other four datasets are OOD.

same setup and framework as proposed in MultiOOD for experiments. More details on the MultiOOD benchmark are in [17].

## D  Implementation Details

For the **Multimodal OOD Segmentation** task, we follow [17] to adopt the fusion framework from PMF [53], modifying it by adding an additional segmentation head to the combined features from the camera and LiDAR streams. We use ResNet-34 [22] as the backbone for the camera stream and SalsaNext [10] for the LiDAR stream. For optimization, we use SGD with Nesterov [41] for the camera stream and Adam [29] for the LiDAR stream. The networks are trained for 50 epochs with a batch size of 4, starting with a learning rate of 0.0005 and with a cosine schedule. To prevent overfitting, we apply various data augmentation techniques, including random horizontal flipping, random scaling, color jitter, 2D random rotation, and random cropping. For hyperparameters, we set $N$ in Feature Mixing to 10 and $\gamma_1$ in loss to 3.0. For A2D, we set $\gamma_2$ to 1.0. For xMUDA, we set $\gamma_2$ to 0.5. For the **Multimodal OOD Detection** task, we conduct experiments across video and optical flow modalities using the MultiOOD benchmark [17]. We use the SlowFast network [20] to encode video data and the SlowFast network's slow-only pathway for optical flow. The models are pre-trained on each dataset's training set using standard cross-entropy loss. The Adam optimizer [29] is employed with a learning rate of 0.0001 and a batch size of 16. For hyperparameters, we set $N$ in Feature Mixing to 512.

## E  Compatible with Cross-modal Training Techniques

Our proposed framework not only supports the basic late-fusion strategy, but is also compatible with advanced cross-modal training techniques that promote interaction across modalities. To demonstrate its versatility, we show how to integrate A2D [17] and xMUDA [26] into the framework.

**Agree-to-Disagree (A2D)**, designed for multimodal OOD detection, aims to amplify the modality prediction discrepancy during training. It assumes additional outputs $\mathbf{O}^c$ and $\mathbf{O}^l$ from each modality. By removing the $c$-th value from $\mathbf{O}^c$ and $\mathbf{O}^l$, A2D derives new prediction probabilities without ground-truth classes, denoted as $\bar{\mathbf{O}}^c$ and $\bar{\mathbf{O}}^l \in \mathbb{R}^{M \times (C-1)}$. A2D then seeks to maximize the discrepancy between $\bar{\mathbf{O}}^c$ and $\bar{\mathbf{O}}^l$, which is defined as:

$$\mathcal{L}_{A2D} = -\frac{1}{M} \sum_{m=1}^{M} D(\bar{\mathbf{O}}_m^c, \bar{\mathbf{O}}_m^l), \tag{31}$$

where $D(\cdot)$ is a distance metric quantifying the similarity between two probability distributions. By integrating A2D into the framework, the final loss function becomes:

$$\mathcal{L} = \mathcal{L}_{foc} + \mathcal{L}_{lov} + \gamma_1 \mathcal{L}_{ent} + \gamma_2 \mathcal{L}_{A2D}. \tag{32}$$

**xMUDA** facilitates cross-modal learning by encouraging information exchange between modalities, allowing them to learn from each other. xMUDA also assumes additional outputs $\mathbf{O}^c$ and $\mathbf{O}^l$ from each modality and define cross-modal loss as:

$$\mathcal{L}_{xM} = \mathbf{D}_{KL}(\mathbf{O}^c || \mathbf{O}) + \mathbf{D}_{KL}(\mathbf{O}^l || \mathbf{O}), \tag{33}$$

where $\mathbf{D}_{KL}$ is the Kullback–Leibler divergence and the final loss function in this case is:

$$\mathcal{L} = \mathcal{L}_{foc} + \mathcal{L}_{lov} + \gamma_1 \mathcal{L}_{ent} + \gamma_2 \mathcal{L}_{xM}. \tag{34}$$

## F  More Ablation Studies

**Hyperparameter Sensitivity.** We evaluate the sensitivity of $\gamma_1$ in the loss function on the SemanticKITTI dataset. Our findings, as illustrated in Fig. 15, demonstrate that training with multimodal outlier generation and optimization consistently outperforms the baseline across all parameter settings. These ablations suggest that our approach is robust and exhibits minimal sensitivity to variations in hyperparameter choices.

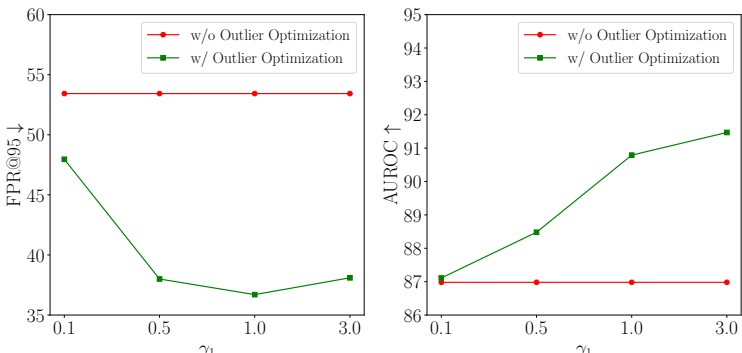

Figure 15: Ablation of $\gamma_1$ in the loss function on SemanticKITTI.

| OOD Scores | FPR@95↓ | AUROC↑ | AUPR↑ |
|---|---|---|---|
| MaxLogit [23] | 31.76 | 92.83 | 61.99 |
| MSP [24] | 32.93 | 91.57 | 51.68 |
| Energy [36] | 32.05 | 92.87 | 64.87 |
| Entropy [13] | 33.00 | 92.50 | 59.11 |
| GEN [37] | 32.63 | 93.00 | 64.43 |

Table 7: Ablation of OOD Scores on SemanticKITTI.

**Impact of Various OOD Scores.** We evaluate the impact of different commonly used OOD scores by replacing the OOD detection module in our framework with MaxLogit [23] (our default), MSP [24], Energy [36], Entropy [13], and GEN [37]. As shown in Appendix F, the FPR@95 and AUROC show minimal fluctuation (less than $2\%$) across different OOD scores, further demonstrating the adaptability of our framework to various design choices.

**Multimodal OOD Detection Results on Kinetics-600.** Tab. 8 presents the multimodal OOD detection results using Kinetics-600 as the ID dataset. Feature Mixing outperforms other outlier generation methods in most cases, achieving the lowest FPR@95 of $54.75\%$ and the highest AUROC of $79.23\%$ on average when using HMDB51 as ID. These results further demonstrate the effectiveness of Feature Mixing in improving OOD detection across diverse tasks and modalities, while maintaining negligible impact on ID accuracy.

**Feature Mixing in Unimodal Settings for OOD Segmentation.** We further extend the unimodal FM variant to the OOD segmentation task using LiDAR-only input. The same strategy used for unimodal classification is directly applied to LiDAR point cloud features. As shown in Tab. 9a, our method achieves competitive performance compared to NP-Mix and baseline methods, demonstrating the scalability of FM to segmentation tasks in unimodal settings with minimal modification.

**Sensitivity of Parameter $N$ on SemanticKITTI.** We further investigate the sensitivity of FM to the mixing parameter $N$ (i.e., the number of swapped feature dimensions) on the SemanticKITTI dataset. As shown in Tab. 9b, FM achieves stable performance across different values of $N$, supporting the robustness of the method. We use $N{=}10$ by default in our main experiments.

## G   Further Discussions on Feature Mixing

**Challenges of extending existing solutions to multimodal setting.** While methods such as Mixup and VOS have shown success in unimodal tasks, directly applying them to multimodal data is non-trivial due to the following key challenges:

(1) *Modality heterogeneity*: Multimodal data often involves inputs with fundamentally different structures and distributions—e.g., dense 2D images, sparse 3D point clouds, temporal audio signals, etc. Pixel-level interpolation techniques like Mixup are inherently incompatible across heterogeneous modalities, making it difficult to define meaningful cross-modal augmentations. Moreover, Mixup's

| Methods | OOD Datasets | | | | | | | | | | ID ACC ↑ |
|---|---|---|---|---|---|---|---|---|---|---|---|
| | HMDB51 | | UCF101 | | EPIC-Kitchens | | HAC | | Average | | |
| | FPR@95↓ | AUROC↑ | FPR@95↓ | AUROC↑ | FPR@95↓ | AUROC↑ | FPR@95↓ | AUROC↑ | FPR@95↓ | AUROC↑ | |
| Baseline | 72.64 | 71.75 | 70.12 | 71.49 | 43.66 | 82.05 | 61.50 | 74.99 | 61.98 | 75.07 | 73.14 |
| Mixup [52] | 66.53 | 70.33 | 68.36 | 69.53 | 39.68 | 86.62 | 56.35 | 77.95 | 57.73 | 76.11 | 73.40 |
| VOS [19] | 65.23 | 71.48 | 68.29 | 73.97 | 38.12 | 86.05 | 56.11 | 79.02 | 56.94 | 77.63 | 73.07 |
| NPOS [48] | 65.72 | 70.93 | 68.29 | 73.97 | 35.13 | 86.78 | 55.89 | 80.49 | 56.26 | 78.04 | 73.49 |
| NP-Mix [17] | 63.27 | 74.17 | 67.20 | 74.50 | 34.07 | 87.49 | 56.69 | 80.20 | 55.31 | 79.09 | 73.67 |
| Feature Mixing (ours) | 62.86 | 74.32 | 67.74 | 74.38 | 33.51 | 87.64 | 54.89 | 80.58 | **54.75** | **79.23** | 73.67 |

Table 8: Multimodal OOD Detection using video and optical flow, with **Kinetics-600** as ID. Energy is used as the OOD score.

| Method | FPR@95↓ | AUROC↑ | AUPR↑ | mIoU↑ |
|---|---|---|---|---|
| Baseline | 47.03 | 85.82 | 36.06 | 59.81 |
| NP-Mix | 48.26 | 86.79 | 45.73 | 59.52 |
| Feature Mixing | **46.40** | **88.81** | **47.67** | **60.04** |

| $N$ | FPR@95↓ | AUROC↑ | AUPR↑ | mIoU↑ |
|---|---|---|---|---|
| 8 | 38.58 | 91.14 | 53.45 | **61.59** |
| 10 | **38.10** | **91.47** | **58.74** | 61.18 |
| 12 | 37.17 | 90.90 | 55.19 | 60.75 |

(a) Unimodal feature mixing for OOD segmentation on SemanticKITTI (LiDAR only).

(b) Ablation of $N$ for OOD segmentation on SemanticKITTI.

Table 9: More ablations on the OOD segmentation task.

random linear blending can produce ambiguous or noisy samples that may lie near or within the in-distribution (ID) manifold, reducing its effectiveness for OOD detection.

(2) *Computational inefficiency*: Methods like VOS and NP-Mix rely on distribution estimation, sampling, or searching in high-dimensional feature spaces, which becomes computationally expensive when extended to multimodal settings—especially for dense prediction tasks such as segmentation, where per-pixel or per-point processing is required.

Our Feature Mixing addresses these challenges by (i) operating entirely in the feature space, where modality-specific structures have already been abstracted, (ii) introducing a lightweight, swap-based perturbation strategy to synthesize outliers efficiently, avoiding costly sampling or searching, (iii) being theoretically grounded (Theorems 1 and 2) and empirically validated to maintain ID-OOD separation while improving efficiency and scalability.

**Clarification on the properties of Feature Mixing and their benefit to OOD detection.** Theoretical results in Theorems 1 and 2 establish two key properties of Feature Mixing: (1) the synthesized outliers have low likelihood under the ID distribution, and (2) their deviation from the ID distribution is bounded. These properties are important for OOD detection and segmentation, as they ensure that *FM produces challenging but plausible outliers that populate the low-density regions near the boundary of the ID distribution*. This supports effective decision boundary regularization and reduces model overconfidence on unseen data. As shown in Fig. 4, the FM-generated outliers are well-separated from the ID features in the embedding space. This separation helps the model to better distinguish between in-distribution and out-of-distribution regions. These visualizations provide intuitive, empirical evidence that our method generates meaningful and bounded outliers, as predicted by our theoretical analysis.

**More details on outlier optimization.** The core idea is that the feature mixing branch generates pseudo-OOD features. We apply an entropy maximization loss in Eq. (5) on these features to explicitly encourage the model to produce uncertain (i.e., high-entropy) predictions for them. This discourages the model from making overconfident predictions on ambiguous or unfamiliar inputs—behavior that is characteristic of OOD samples. In contrast, the classification loss encourages low-entropy (i.e., high-confidence) predictions on in-distribution (ID) samples. Together, these complementary objectives train the model to develop a confidence-based decision boundary: ID samples are associated with confident predictions, while pseudo-OOD samples—introduced through feature mixing—are explicitly pushed toward uncertain predictions. As shown in Fig. 2, this training objective increases the separation in predictive confidence between ID and OOD inputs, which is crucial for effective OOD detection.

