# OpenReview forum: "Extremely Simple Multimodal Outlier Synthesis for Out-of-Distribution Detection and Segmentation"
_NeurIPS.cc/2025/Conference — NeurIPS 2025 poster_

### Official Review · Reviewer_4Bsi · 2025-06-16

**Clarity:** 3
**Significance:** 2
**Originality:** 3
**Rating:** 5
**Confidence:** 3

**Summary:**

This work proposes a new outlier synthesis method for multimodal OOD detection and segmentation. The idea is to randomly swap the elements at certain dimension of the features between the two modalities. Theoretical proofs are presented to indicate that the synthesized outliers may share similar characteristics to the real ones in deployment environment. Empirical experiments support the two major claims of this work, that the proposed method leads to improved performance while being much more efficient in outlier synthesis (in terms of runtime).

**Questions:**

Please see Weaknesses/Questions/Suggestions.

**Ethical Concerns:**

["NO or VERY MINOR ethics concerns only"]

**Final Justification:**

My questions are well-addressed by the rebuttal, where I find the additional results under three-modality and single-modality scenarios quite helpful, demonstrating the great flexibility of the proposed approach. I'm thus raising my score from 4 (pre-rebuttal) to 5 (final assessment).

**Limitations:**

It is claimed in the paper checklist that "In the Appendix, we discussed limitations and future work." Yet I didn't see such discussion in the appendix. It would be helpful to discuss whether scaling to more modalities would be an limitation of the proposed method.

**Quality:**

3

**Strengths And Weaknesses:**

### Strengths

1. The two major claims (improved performance and efficiency) are supported well.

### Weaknesses/Questions/Suggestions

1. The swapping mechanism seems to be specific for cases where there are strictly two modalities. What if there are more modalities? How are the features going to be swapped?
2. It's good to have theoretical proofs in section 3.3, yet the two conclusions derived from the proofs (the "two key properties" at line 179) are somewhat ambiguous in terms of how they are going to benefit OOD detection. Instead, I think a more straightforward approach is to visualize real outliers together with synthesized outliers in Figure 4 and see if they have large overlap. If so, then it is a strong evidence that the synthesized outliers could be useful in regularizing the model's outputs against real outliers (since they tend to overlap in model feature space).
3. In appendix line 8 Eq. (1) and (2), the mask is the same for either modality's features (with the same notation $\mathcal{M}$). This is not consistent with what's being shown in Figure 3 and Algorithm 1, where there are obviously two different masks for each modality.

---

> ### Author Rebuttal · Authors · 2025-07-30
>
> Thanks for your insightful reviews and great support of our paper! We provide the responses to your questions as follows:
> >**Q1**: What if there are more modalities? How are the features going to be swapped?
>
> **A1**: Thanks for your suggestion! To further demonstrate the generality of our approach, we conducted an additional experiment on a **three-modality setting involving video, optical flow, and audio** for OOD detection on the EPIC-Kitchens dataset. We extend Feature Mixing to this setting by randomly selecting a subset of N feature dimensions from each modality and swapping them cyclically: video->audio, audio->optical flow, and optical flow->video, thus creating outlier features for all three modalities. As shown in the table below, Feature Mixing consistently outperforms NP-Mix across all metrics in this three-modality setting (evaluated using the Energy score). These results further support the robustness and versatility of our method.
>
> ||FPR95$\downarrow$|AUROC$\uparrow$|ACC$\uparrow$|
> |-|-|-|-|
> |Baseline |69.22| 72.39| 73.13|
> |NP-Mix|62.69| 74.95 |71.46|
> |Feature Mixing|61.38 |77.23 |73.69|
>
> Importantly, FM is inherently **modality-agnostic**: it operates in the feature space without relying on assumptions about the input structure, enabling straightforward application to diverse and heterogeneous modality combinations. We further extend FM to the **single-modality setting** by randomly selecting N dimensions from the first and second halves of the feature embedding and swapping them within the same modality. This enables outlier synthesis without relying on a second modality. We evaluate this unimodal variant on OOD detection using HMDB51 (video modality). As shown below, our unimodal FM achieves strong performance, outperforming the best NP-Mix baseline, which validates the applicability of FM beyond multimodal settings.
> ||FPR95$\downarrow$|AUROC$\uparrow$|ACC$\uparrow$|
> |-|-|-|-|
> |Baseline |64.05| 83.14 |86.93|
> |NP-Mix|62.53 |83.62| 87.15|
> |Feature Mixing|57.30| 84.41 |88.89|
>
> ___
> >**Q2**: Clarification of the "two key properties" and their benefit to OOD detection
>
> **A2**: Thank you for the thoughtful suggestion. Theoretical results in Theorems 1 and 2 establish two key properties of Feature Mixing (FM): (1) the synthesized outliers have low likelihood under the ID distribution, and (2) their deviation from the ID distribution is bounded. These properties are important for OOD detection and segmentation, as they ensure that **FM produces challenging but plausible outliers that populate the low-density regions near the boundary of the ID distribution**. This supports effective decision boundary regularization and reduces model overconfidence on unseen data.
>
> As shown in Figure 4, the FM-generated outliers are well-separated from the ID features in the embedding space. This separation helps the model to better distinguish between in-distribution and out-of-distribution regions. These visualizations provide intuitive, empirical evidence that our method generates meaningful and bounded outliers, as predicted by our theoretical analysis.
>
> In response to your suggestion, we have additionally visualized real outliers alongside FM-synthesized outliers, and we observe a significant degree of overlap between them. This provides empirical evidence that the synthesized outliers are representative of real OOD samples, which supports their effectiveness in regularizing the model. Interestingly, **FM-synthesized outliers also extend into other low-likelihood regions not covered by the real OOD samples, further increasing the diversity and coverage of the synthesized data**.
> ___
> >**Q3**: the mask is not consistent
>
> **A3**: Thank you for pointing this out, and we apologize for the confusion.
> In Appendix, we used the same mask notation for both modalities in Equations (1) and (2) for notational simplicity in the theoretical analysis. However, as correctly illustrated in Figure 3 and Algorithm 1, we do use two distinct masks—one for each modality—in the actual implementation.
>
> This simplification in notation **does not affect the validity of the theoretical results**. For example, the expectations computed in Equations (3) and (4) in appendix remain the same regardless of whether the two modalities share the same mask or not, since the proofs rely on the marginal distributions rather than the specific mask values.
>
> We will correct the notation in the final version of the paper to accurately reflect the use of distinct masks and to ensure consistency throughout the manuscript.
> ___
> >**Q4**: discussion on limitations
>
> **A4**: Sorry for the confusion. We discuss the limitations and future work here and will include them in the final version of the paper.
>
> Feature Mixing uses a random selection of feature dimensions to swap between modalities. While this is highly efficient and theoretically grounded, it may not always target the most informative feature regions for outlier synthesis. Future work could explore **adaptive or learnable selection mechanisms** that dynamically identify features that maximize OOD separability.
> ___
> We sincerely thank the reviewer again for the constructive feedback. We will incorporate all clarifications and updates from this rebuttal into the final version to make the paper clearer. If there are any additional questions or concerns, we would be more than happy to further address them.

---

### Official Review · Reviewer_rcrp · 2025-06-20

**Clarity:** 3
**Significance:** 3
**Originality:** 3
**Rating:** 5
**Confidence:** 4

**Summary:**

The paper introduces a simple but effective outlier synthesis procedure to enable better multimodal out-of-distribution detection (OOD). This procedure works by selecting random N indices for each modality feature map and swapping them. This guarantees that the outliers lie in a low-likelihood region and do not extremely deviate from the original features. Both of these claims are also theoretically proven. These outliers are then used in the entropy maximisation learning objective to prevent overconfidence for OOD samples. The procedure is evaluated for multimodal OOD segmentation and multimodal OOD classification. It achieves solid SOTA results in both scenarios.

**Questions:**

Besides the questions already included in the weaknesses section, I here list two that will have the most impact on my final opinion:
- How does the parameter N affect the segmentation performance?
- How hard is it to extend this to single modality? If not too hard to modify, what’s the performance like for segmentation with just RGB data used for both outlier synthesis and inference?

**Ethical Concerns:**

["NO or VERY MINOR ethics concerns only"]

**Final Justification:**

I strongly believe this paper **should be accepted**.
The authors have addressed all my concerns. Most were resolved in rebuttal, and a final question was resolved in the discussion.
The paper is clearly written and easy to read. The problem is nicely stated, and the solution is elegant and well-explained. The approach outperforms related methods on various benchmarks. The authors also introduce a new benchmark, which will be beneficial for the field.

**Limitations:**

yes

**Paper Formatting Concerns:**

/

**Quality:**

3

**Strengths And Weaknesses:**

Strengths:
- The paper is nicely written and easy to follow.
- I like the simplicity of the proposed method; it addresses the issue without overcomplicating the whole process. This, combined with its applicability to related frameworks, makes the contribution strong, despite its simplicity.
- The results achieved are solid and convincing.

Weaknesses:
- You say you set the OOD classes in SemanticKITTI and nuScenes to void and ignore them. Have you considered whether the model learns to implicitly detect these classes due to this decision?
- While the method is designed for multimodal OOD, I would be interested in seeing how it performs in the monomodal scenario. Would the method still work if you just randomly shuffled N features in a single modality?
- You claim that you repeat each training run with 3 random seeds and report the mean. Why not include the standard deviation, at least in the appendix?
- It is not entirely clear how the method works for segmentation. Do you perform this mixing for each spatial location in feature maps? Are the indices distinct for each location? I would also like to see the ablation study of the parameter N for segmentation, not just detection.

Minor:
- You say that the work has no societal impact, yet I think that OOD detection for self-driving cars has some impact. I think it would be appropriate to include some discussion in the appendix.

---

> ### Author Rebuttal · Authors · 2025-07-30
>
> Thanks for your insightful reviews and great support of our paper! We provide the responses to your questions as follows:
> >**Q1**: whether the model learns to implicitly detect these classes due to this decision
>
> **A1**: Thank you for your insightful comment. We follow the standard protocol in prior OOD segmentation works (e.g., [1]), where certain classes are assigned the void label and are excluded from the loss computation during training. This setup is commonly used due to the absence of publicly available 3D OOD segmentation datasets.
>
> However, as you correctly pointed out, this approach introduces the risk that the model may implicitly learn to detect these void-labeled classes, since their input features are still processed by the network. To address this limitation and eliminate the possibility of such implicit supervision, we **introduce the CARLA-OOD dataset, a new benchmark for multimodal OOD segmentation**. In CARLA-OOD, **OOD classes are strictly held out from the training data and only appear during testing, ensuring a clean evaluation protocol without any data leakage**.
>
> We believe this new dataset provides a more rigorous and realistic testbed for multimodal OOD segmentation in autonomous driving.
>
> [1] Cen et al., Open-world Semantic Segmentation for LIDAR Point Clouds, ECCV 2022
> ___
> >**Q2**: How the method works for segmentation? the ablation study of the parameter N for segmentation?
>
> **A2**: Thank you for the helpful questions, and we apologize for the lack of clarity.
> For segmentation, we first project the LiDAR point cloud onto the image plane, following the approach in [1], to obtain spatially aligned pairs of LiDAR points and image pixels. Feature mixing is then performed at each valid pixel location—i.e., only where a corresponding LiDAR point exists—by mixing the aligned image and point features. Pixels without projected points are excluded from the mixing.
>
> We have now also included the ablation study of the parameter
> N for segmentation on SemanticKITTI, as shown in the table below. We use N=10 by default, and the results show that the performance is **robust across different values of N**.
>
> [1] Zhuang et al., Perception-aware Multi-sensor Fusion for 3D LiDAR Semantic Segmentation, ICCV 2021
>
> ||FPR95$\downarrow$|AUROC$\uparrow$|AUPR$\uparrow$|mIoU$\uparrow$|
> |-|-|-|-|-|
> |8| 38.58 |91.14 |53.45| 61.59|
> |10| 38.10| 91.47 |58.74| 61.18|
> |12 |37.17 |90.90 |55.19| 60.75|
> ___
> >**Q3**: Feature Mixing on a single modality
>
> **A3**: Thank you for the insightful suggestion. We explored extending Feature Mixing to the single-modality setting by **randomly selecting N dimensions from the first and second halves of the feature embedding and swapping them within the same modality**. This approach enables outlier synthesis without relying on a second modality. We first evaluate this unimodal variant on the OOD detection task using the HMDB51 dataset (video modality). As shown in the table below, our **unimodal FM achieves strong performance**, outperforming the best NP-Mix baseline, which validates the applicability of our method beyond multimodal settings.
>
> ||FPR95$\downarrow$|AUROC$\uparrow$|ACC$\uparrow$|
> |-|-|-|-|
> |Baseline |64.05| 83.14 |86.93|
> |NP-Mix|62.53 |83.62| 87.15|
> |Feature Mixing|57.30| 84.41 |88.89|
>
> We then apply the same unimodal FM strategy to the OOD segmentation task on SemanticKITTI using only LiDAR data. The results, also provided in the table below, demonstrate that our method **scales effectively to segmentation and maintains competitive performance in the single-modality case**. These results suggest that FM is flexible and generalizable, and can be extended to unimodal settings with minimal modifications.
>
> ||FPR95$\downarrow$|AUROC$\uparrow$|AUPR$\uparrow$|mIoU$\uparrow$|
> |-|-|-|-|-|
> |Baseline|47.03 |85.82| 36.06| 59.81|
> |NP-Mix|48.26| 86.79 |45.73| 59.52|
> |Feature Mixing|46.40 |88.81 |47.67 |60.04|
>
> To further demonstrate the generality of our approach, we conducted an additional experiment on a **three-modality setting involving video, optical flow, and audio** for OOD detection on the EPIC-Kitchens dataset. We extend Feature Mixing to this setting by randomly selecting a subset of N feature dimensions from each modality and swapping them cyclically: video->audio, audio->optical flow, and optical flow->video, thus creating outlier features for all three modalities. As shown in the table below, Feature Mixing consistently outperforms NP-Mix across all metrics in this three-modality setting (evaluated using the Energy score). These results further support the robustness and versatility of our method.
>
> ||FPR95$\downarrow$|AUROC$\uparrow$|ACC$\uparrow$|
> |-|-|-|-|
> |Baseline |69.22| 72.39| 73.13|
> |NP-Mix|62.69| 74.95 |71.46|
> |Feature Mixing|61.38 |77.23 |73.69|
> ___
> >**Q4**: standard deviations
>
> **A4**: Thank you for pointing this out, and we apologize for the confusion. To maintain clarity and readability in the main paper, especially in longer tables such as Table 1 and Table 2, we chose to report only the mean values, as including standard deviations would have made the tables overly cluttered. That said, we agree that reporting variance is important for assessing the reliability of the results. We confirm that each experiment was run with three different random seeds, and we will include the standard deviations in the appendix in the final version of the paper for completeness.
> ___
> >**Q5**: Societal impact
>
> **A5**: Thank you for your comment. We agree that OOD detection for self-driving cars has important societal implications. In our work, Feature Mixing advances multimodal OOD detection and segmentation by enabling more effective and efficient outlier exposure during training. This contributes directly to the reliability and safety of autonomous systems, including self-driving cars, by helping models handle unfamiliar or unexpected scenarios in open-world environments.
>
> Beyond autonomous driving, the method also has broader societal impact in other safety-critical domains such as healthcare and security, where robustness to unknown or anomalous data is crucial. By improving model reliability in the presence of distribution shifts, our approach **supports the deployment of AI systems in dynamic, high-stakes environments where failure could have significant consequences**.
>
> We will include a discussion of these points in the appendix as suggested.
> ___
> We sincerely thank the reviewer again for the constructive feedback. We will incorporate all clarifications and updates from this rebuttal into the final version to make the paper clearer. If there are any additional questions or concerns, we would be more than happy to further address them.

---

> > ### Comment · Reviewer_rcrp · 2025-08-01
> >
> > Thank you for the detailed rebuttal. It addresses most of my concerns.
> >
> > Great to see that the same modality mixing also works. I think this strengthens the contribution and should be included in the final version, even if in the appendix.
> >
> > Thanks for the clarification on segmentation implementation. Please also add this to the implementation details in the appendix.
> > I also appreciate the ablation on segmentation N, the results seem good. I’m wondering why you use N=10, while in classification the N is 512. What is the full feature dimension in case of segmentation and classification? I think this could also be included in the implementation details.
> > If the full feature dimension in segmentation is much larger than 10, why choose such a low value?

---

> > > ### Author Response · Authors · 2025-08-01
> > >
> > > Dear Reviewer rcrp,
> > >
> > > Thank you for your thoughtful feedback. We're glad to hear that our previous response addressed most of your concerns. Regarding your question about the choice of N=10 in segmentation versus N=512 in classification: Yes, the full feature dimensions differ significantly between the two tasks due to architectural differences. Specifically:
> > >
> > > For classification, we extract features before the final classification layer. The feature dimension is 2304 for video and 2048 for optical flow.
> > >
> > > For segmentation, we extract features after the decoder and before the final segmentation head. The feature dimension is much smaller: 16 for images and 32 for LiDAR point clouds.
> > >
> > > This explains the choice of a smaller N in the segmentation task. We will include these implementation details, along with the new results on unimodal mixing, in the final version of the paper. Please don’t hesitate to let us know if you have any further questions and we’re happy to clarify.

---

> > > > ### Comment · Reviewer_rcrp · 2025-08-01
> > > > **Ready for acceptance**
> > > >
> > > > This addresses all of my requests. Thank you for your swift reply. Overall, I think the contribution is good with these final things clarified for the final version. In line with my rating, **I suggest that the paper should be accepted**.

---

> > > > > ### Author Response · Authors · 2025-08-01
> > > > >
> > > > > Dear Reviewer rcrp,
> > > > >
> > > > > Thank you very much for your encouraging feedback and for taking the time to carefully review our work. We’re glad that our responses have addressed your concerns, and we greatly appreciate your positive evaluation and recommendation for acceptance. Your comments have helped us improve the clarity and completeness of our paper.

---

### Official Review · Reviewer_GQ4N · 2025-07-03

**Clarity:** 2
**Significance:** 3
**Originality:** 3
**Rating:** 4
**Confidence:** 2

**Summary:**

This paper propose Feature Mix, an OOD feature synthesizing method that allows model learns better to distinguish ID and OOD observations. The motivation of this paper is to improve the perception task performance by augment training with synthetic OOD features. Given multimodal input (image, LiDAR, or an agnostic modality), and their encoders, the stochasticity are added to the embeddings, which forms an OOD branch in the  neural network. The output of this OOD branch is being optimized, concretely by maximizing the embedding entropy, to help the network better distinguish the ID and OOD samples. Image segmentation and detection tasks on autonomous tasks based on proposed method are studied to validate the efficiency of proposed method.

**Questions:**

I thinks I mentioned most of the suggestion above:
1. Methodology might need some reorganization to help readers better comprehend. Especially the part on page 6
2. want to see more support around Figure 4 in paper and Figure 1 in appendix other than proofs that show how the proposed simple method is correct.

**Ethical Concerns:**

["NO or VERY MINOR ethics concerns only"]

**Final Justification:**

I would like to thank authors for rebuttal and my scores are remained.

**Limitations:**

I tried to find it in appendix as mentioned but it seems missing?

**Paper Formatting Concerns:**

no.

**Quality:**

3

**Strengths And Weaknesses:**

Strength

* The paper is overall well written and organized. Problem is clearly defined.

* Theorems, methods, proofs and experiments on real application data are in concert with each other, making it a self-containing work.

* I like the fact that the proposed idea is indeed extremely simple and authors went through proofs of that, proving the effectiveness and bound (appendix). I went through the appendix and proofs looks reasonable to me, however I’m not a mathematical guy so I will lower down my confidence rating accordingly. (suggestions below for broader audience in case they are not into proofs)

Weakness

* Given the proposed simple feature mixing strategy, how the mixed branch is optimized is the center of this contribution. However, the explanation around this is not sufficient. It is hard to understand why maximizing the entropy of mixed OOD feature can better differentiate the OOD and ID samples. It is mentioned between line 208 and 213 and I get directed to eq.5.  I guess the narrative flow here might need some revision.

* The 2 theorems proposed in the paper need more direct experimental and visualization support. Figure 4 in the main paper and Figure 1 in the appendix are good, but I would like to know how the data visualized in these figures are generated. Are they 2D or low-dimension toy examples, or are they real world data with reduced dimensions? Also in these figures why the ID data, as shown as purple dots, are not identical? I assumes that the comparison should be conducted on the same ID features as baselines.

* In the table 1 and table 2, (and maybe table 3 and table 4 as well), please use consistent bold text to indicate the best entry in each comparisons. Otherwise they are hard to tell which is better.

---

> ### Author Rebuttal · Authors · 2025-07-30
>
> Thanks for your insightful reviews and great support of our paper! We provide the responses to your questions as follows:
> >**Q1**: how the mixed branch is optimized? need some reorganization to help readers better comprehend
>
> **A1**: Thank you for the insightful comment. We agree that a clearer explanation of how the entropy maximization loss optimizes the mixed branch is important.
> The core idea is that the mixed branch generates pseudo-OOD features through feature mixing. We apply an entropy maximization loss (Eq. 5) on these features to explicitly **encourage the model to produce uncertain (i.e., high-entropy) predictions** for them. This discourages the model from making overconfident predictions on ambiguous or unfamiliar inputs—behavior that is characteristic of OOD samples.
>
> In contrast, the classification loss encourages low-entropy (i.e., high-confidence) predictions on in-distribution (ID) samples. Together, these complementary objectives train the model to develop a confidence-based decision boundary: **ID samples are associated with confident predictions, while pseudo-OOD samples—introduced through feature mixing—are explicitly pushed toward uncertain predictions.**
>
> As shown in Figure 2, this training objective **increases the separation in predictive confidence between ID and OOD inputs**, which is crucial for effective OOD detection.
> We appreciate your feedback and will revise the narrative between lines 208 and 213 to better highlight this motivation and improve the clarity around the optimization of the mixed branch.
> ___
> >**Q2**: support around Figure 4 in paper and Figure 1 in appendix
>
> **A2**: Thank you for the thoughtful question. We apologize for the confusion. The visualizations in Figure 4 (main paper) and Figure 1 (appendix) are based on **real-world data from the HMDB51** dataset—not on toy or synthetic examples. Specifically, we apply our feature mixing strategy to generate pseudo-OOD features, and then use t-SNE to project the high-dimensional features into 2D for visualization.
>
> Regarding the ID features (purple dots): they are indeed **identical across all methods**. The apparent differences in their positions are due to the stochastic nature of t-SNE, which may project the same high-dimensional points differently depending on the surrounding context (i.e., the presence of different OOD samples). Since t-SNE emphasizes preserving local neighborhood structures rather than global geometry, the introduction of different OOD features can cause slight shifts in the 2D projections of the ID features as well.
>
> To further clarify how these visualizations support our theoretical claims: Theorems 1 and 2 establish that feature mixing generates samples that lie outside the ID manifold (i.e., are valid outliers), yet remain bounded in their deviation. This is empirically reflected in Figure 4(d): the pseudo-OOD features are clearly separated from the ID cluster (supporting Theorem 1), while remaining relatively close rather than scattered far away (supporting Theorem 2). For Figure 1 (appendix), we prove in Eq. 6 (appendix) that the shifts of outliers from the ID distribution are proportional to N. Therefore, a large N promotes outliers to lie in low-likelihood regions of the ID feature distribution.
>
> Together, these visualizations provide intuitive, empirical evidence that our method generates meaningful and bounded outliers, as predicted by our theoretical analysis. We will revise the text to clarify how the figures are generated and better connect the visualizations to Theorems 1 and 2.
> ___
> >**Q3**: Consistent bold text in tables
>
> **A3**: Thank you for pointing this out. We acknowledge the inconsistency in highlighting the best-performing entries in Tables 1–4. In the final version, we will ensure that all tables consistently use bold text to indicate the best results in each comparison. This will significantly improve readability and make performance differences easier to interpret.
> ___
> >**Q4**: discussion on limitations
>
> **A4**: Sorry for the confusion. We discuss the limitations and future work here and will include them in the final version of the paper.
>
> Feature Mixing uses a random selection of feature dimensions to swap between modalities. While this is highly efficient and theoretically grounded, it may not always target the most informative feature regions for outlier synthesis. Future work could explore **adaptive or learnable selection mechanisms** that dynamically identify features that maximize OOD separability.
> ___
> We sincerely thank the reviewer again for the constructive feedback. We will incorporate all clarifications and updates from this rebuttal into the final version to make the paper clearer. If there are any additional questions or concerns, we would be more than happy to further address them.

---

> > ### Comment · Reviewer_GQ4N · 2025-08-08
> >
> > I would like to thank authors for the rebuttal and answer my questions. I have no further concerns.

---

### Official Review · Reviewer_an3N · 2025-07-03

**Clarity:** 3
**Significance:** 2
**Originality:** 2
**Rating:** 4
**Confidence:** 4

**Summary:**

This paper addresses the problem of outlier synthesis for out-of-distribution (OOD) detection. The key observation is that neural networks often produce high-confidence predictions for OOD samples, and outlier synthesis can help mitigate this overconfidence. The authors point out that most existing outlier synthesis methods are designed for unimodal settings and have limitations when applied to multimodal data. To address this, the paper proposes a new feature mixing strategy tailored for the multimodal setting. Specifically, given in-distribution features from two modalities, the method randomly swaps a subset of N feature dimensions between them to generate synthetic multimodal outliers. Experimental results demonstrate both the efficiency and effectiveness of the proposed approach.

**Questions:**

1. In line 150, the authors state that Mixup may generate noisy samples that still lie within the ID distribution. Further clarification is needed to support the claim that the proposed method avoids this limitation. While visualizing features directly may be challenging, some form of intuitive or empirical analysis would strengthen the argument.
2. What are the potential challenges or limitations in directly extending Mixup or VOS to multimodal settings? A brief discussion would help highlight the motivation for the proposed approach.
3. How does the proposed feature mixing strategy differ from the dynamic mix-and-match perturbation introduced in “A Stochastic Conditioning Scheme for Diverse Human Motion Prediction” (CVPR 2020)? A comparison would help clarify the novelty of the method.
4. Given that the framework operates in feature space, it would be valuable to include a decoder or reconstruction pipeline that maps the mixed features back to image space. This could provide further interpretability and insight into the nature of the synthesized outliers.

**Ethical Concerns:**

["NO or VERY MINOR ethics concerns only"]

**Final Justification:**

This paper addresses the problem of outlier synthesis for out-of-distribution (OOD) detection. My initial concerns were its relation to existing work and the challenges of extending from the unimodal to the multimodal setting. The authors have provided extensive evidence addressing these points, and I therefore raise my rating to a borderline accept.

**Limitations:**

yes

**Paper Formatting Concerns:**

There exists no paper formatting concerns.

**Quality:**

3

**Strengths And Weaknesses:**

Strengths:
1. The proposed multimodal feature mixing strategy for outlier synthesis is intuitive and effective. By operating in the feature space rather than the pixel space, it enables efficient generation of synthetic multimodal outliers.

Weaknesses:
1. While the feature mixing approach demonstrates good empirical performance, its effectiveness across different modality types is not thoroughly analyzed. A deeper investigation into how the method behaves with varying modality combinations would help better assess its generalizability and contribution.
2. The use of entropy maximization for OOD calibration is not novel. For example, similar ideas have been explored in "Entropy Maximization and Meta Classification for Out-of-Distribution Detection in Semantic Segmentation" (ICCV 2021). A clearer positioning of the proposed method relative to such prior work would strengthen the paper.
3. The challenges of extending existing unimodal feature mixing solutions to multimodal setting are not clearly explained.

---

> ### Author Rebuttal · Authors · 2025-07-30
>
> Thanks for your insightful reviews and valuable comments on our paper! We provide the responses to your questions as follows:
> >**Q1**: how the method behaves with varying modality combinations
>
> **A1**: Thank you for raising this important point. We agree that analyzing the method's behavior across diverse modality combinations is crucial for assessing its generalizability. In our current experiments, **we already evaluated Feature Mixing (FM) on substantially different modality pairs**, such as point cloud + image for multimodal OOD segmentation (Table 1), as well as video + optical flow for multimodal OOD detection (Table 2). To further demonstrate the generality, we conducted an additional experiment on a **three-modality setting involving video, optical flow, and audio** for OOD detection on EPIC-Kitchens dataset. We extend FM to this setting by randomly selecting a subset of N feature dimensions from each modality and swapping them cyclically: video->audio, audio->optical flow, and optical flow->video, thus creating outlier features for all three modalities. As shown below, FM consistently outperforms NP-Mix across all metrics in this three-modality setting. These results further support the robustness and versatility of our method.
>
> ||FPR95$\downarrow$|AUROC$\uparrow$|ACC$\uparrow$|
> |-|-|-|-|
> |Baseline |69.22| 72.39| 73.13|
> |NP-Mix|62.69| 74.95 |71.46|
> |Feature Mixing|61.38 |77.23 |73.69|
>
> Importantly, FM is inherently **modality-agnostic**: it operates in the feature space without relying on assumptions about the input structure, enabling straightforward application to diverse and heterogeneous modality combinations. We further extend FM to the **single-modality setting** by randomly selecting N dimensions from the first and second halves of the feature embedding and swapping them within the same modality. This enables outlier synthesis without relying on a second modality. We evaluate this unimodal variant on OOD detection using HMDB51 (with video modality). As shown below, our unimodal FM achieves strong performance, outperforming the best NP-Mix baseline, which validates the applicability of FM beyond multimodal settings.
> ||FPR95$\downarrow$|AUROC$\uparrow$|ACC$\uparrow$|
> |-|-|-|-|
> |Baseline |64.05| 83.14 |86.93|
> |NP-Mix|62.53 |83.62| 87.15|
> |Feature Mixing|57.30| 84.41 |88.89|
> ___
> >**Q2**: positioning relative to prior work on entropy maximization
>
> **A2**:  Thank you for the helpful comment. We would like to clarify that entropy maximization is not the main contribution of our work—it serves as a lightweight auxiliary component for outlier training. The **core novelty of our approach lies in multimodal outlier synthesis via Feature Mixing**, which addresses key limitations of prior methods, particularly in terms of scalability and computational efficiency in multimodal settings. While entropy-based techniques (e.g., ICCV 2021 work) have been explored in the unimodal segmentation context, they typically rely on external OOD datasets for training and do not generalize well to multimodal scenarios. In contrast, our method performs online outlier synthesis using only in-distribution data, eliminating the need for external data, and integrates entropy regularization as a self-contained, plug-in loss. Additionally, our method is significantly more efficient—achieving up to 370× speed-up over prior approaches. We will revise the paper to clarify this positioning and more clearly distinguish our contributions from prior work.
> ___
> >**Q3**: challenges of extending existing solutions to multimodal setting
>
> **A3**: We agree that clarifying the challenges of extending unimodal methods to the multimodal setting is important for highlighting our motivation. While methods such as Mixup and VOS have shown success in unimodal tasks, directly applying them to multimodal data is non-trivial due to the following key challenges:
>
> **(1) Modality heterogeneity:** Multimodal data often involves inputs with fundamentally different structures and distributions—e.g., dense 2D images, sparse 3D point clouds, temporal audio signals, etc. Pixel-level interpolation techniques like Mixup are inherently incompatible across heterogeneous modalities, making it difficult to define meaningful cross-modal augmentations. Moreover, Mixup's random linear blending can produce ambiguous or noisy samples that may lie near or within the in-distribution (ID) manifold, reducing its effectiveness for OOD detection.
>
> **(2) Computational inefficiency:** Methods like VOS and NP-Mix rely on distribution estimation, sampling, or searching in high-dimensional feature spaces, which becomes computationally expensive when extended to multimodal settings—especially for dense prediction tasks such as segmentation, where per-pixel or per-point processing is required.
>
> Our Feature Mixing addresses these challenges by (i) operating entirely in the feature space, where modality-specific structures have already been abstracted, (ii) introducing a lightweight, swap-based perturbation strategy to synthesize outliers efficiently, avoiding costly sampling or searching, (iii) being theoretically grounded (Theorems 1 and 2) and empirically validated to maintain ID-OOD separation while improving efficiency and scalability.
>
> We will revise the paper to include this discussion and better highlight the motivation and advantages of our approach.
> ___
> >**Q4**: intuitive or empirical analysis on Feature Mixing
>
> **A4**: Thank you for the helpful comment. We provide both **theoretical and empirical evidence** to support the claim that our proposed method generates true outliers without injecting noise.
>
> Theoretically, Theorems 1 and 2 in the paper prove that samples generated via Feature Mixing have a low likelihood of residing within the in-distribution (ID) manifold, satisfying the criteria for true outliers. In contrast to interpolation-based methods like Mixup, our approach perturbs features in a structured manner, which helps avoid overlapping with the ID region.
>
> Empirically, we visualize the synthesized outliers using t-SNE on HMDB51 dataset in Figure 4. As shown, Mixup tends to generate noisy samples that remain close to or within the ID feature cluster. In contrast, Feature Mixing produces outliers that are more clearly separated from the ID distribution, effectively expanding the boundary of the embedding space.
> ___
> >**Q5**: comparison with mix-and-match
>
> **A5**: While our Feature Mixing and the Mix-and-Match strategy both involve perturbations in feature space, they differ fundamentally in **objective, mechanism, and application domain**:
>
> **1. Different Objectives**: The primary goal of mix-and-match is to encourage diversity within the data distribution for stochastic human motion prediction. In contrast, our Feature Mixing is designed to synthesize OOD samples for the explicit purpose of improving OOD detection and segmentation. We aim to create features that lie in low-likelihood regions of the in-distribution data, rather than generating plausible variations of it.
>
> **2. Different Mechanisms**: Mix-and-match perturbs a unimodal hidden state with random noise. Our Feature Mixing, however, operates in a multimodal setting by swapping feature dimensions between two different modalities (e.g., images and point clouds). This cross-modal exchange is a core element of our approach and is fundamentally different from adding noise to a single modality's representation.
>
> **3. Different Applications**: Mix-and-match is applied to generative modeling for sequence prediction. Our method is designed for discriminative models in the context of OOD detection.
>
> In summary, while both methods involve a form of feature perturbation, our Feature Mixing is a novel approach for multimodal OOD synthesis, distinct from the unimodal, diversity-focused mix-and-match perturbation.
> ___
> >**Q6**: include a decoder or reconstruction pipeline
>
> **A6**: Thank you for the thoughtful suggestion. We agree that incorporating a decoder or reconstruction pipeline from feature space back to image space could enhance interpretability by providing visual intuition about the synthesized outliers. In this work, our primary focus is on efficient multimodal outlier generation in the feature space, which avoids the complexity and overhead of generative modeling. As shown in Figure 4, we already provide t-SNE visualizations to empirically demonstrate that the outliers generated by Feature Mixing are well-separated from the in-distribution data, supporting their OOD nature.
>
> We **explored the feasibility of feature-to-image inversion in a controlled setting using a text-to-image diffusion model** (Stable Diffusion 3 medium). Specifically, we used the text prompt "a photo of a dog" and extracted embeddings for the text. We then applied Feature Mixing by randomly selecting N dimensions from the first and second halves of the text feature embeddings and swapping them. Finally, we used the resulting text embedding as input to Stable Diffusion 3 to generate an image. We found that the generated images shifted away from the dog concept, indicating that the perturbed embedding moved outside the in-distribution manifold. This serves as initial evidence that Feature Mixing does in fact produce semantically meaningful outliers in embedding space. For multimodal scenarios, however, such as video or 3D point clouds, constructing reliable decoders is substantially more difficult due to data scarcity and the complexity of generative modeling across heterogeneous modalities. We therefore consider feature-to-modality inversion for complex modalities a valuable and promising direction for future work.
> ___
> We sincerely thank the reviewer again for the constructive feedback. We will incorporate all clarifications and updates from this rebuttal into the final version to make the paper clearer. If there are any additional questions or concerns, we would be more than happy to further address them.

---

> > ### Comment · Reviewer_an3N · 2025-08-05
> >
> > The authors have provided a thorough response that addresses all of my concerns. I would therefore like to raise my final rating.

---

> > > ### Author Response · Authors · 2025-08-05
> > > **Thanks for recognizing our efforts and your willingness to raise the final rating!**
> > >
> > > Dear Reviewer an3N,
> > >
> > > We are glad to hear that we have addressed all your concerns and that you would like to raise the final rating! Thanks for spending a significant amount of time on our submission and giving lots of valuable and insightful suggestions, which make our paper even stronger! We will include all added experiments and discussions in the final paper for better clarification.

---

> ### Author Response · Authors · 2025-08-05
>
> Dear Reviewer an3N,
>
> We’ve carefully addressed all your insightful comments. Your feedback has been invaluable in improving the quality of our work, and we greatly appreciate your thoughtful review.
>
> As the discussion period is nearing its end, we wanted to ensure you’ve had the opportunity to review our responses. Please let us know if there are any remaining questions or concerns we can address further.
>
> Thank you once again for your time and consideration.

---

### Note · Authors · 2025-08-12

We deeply appreciate the reviewers and ACs for their careful evaluation and constructive feedback. The discussion period was highly valuable, and we are pleased to have thoroughly addressed all questions and concerns raised by the reviewers.

Our work introduces Feature Mixing, an **extremely simple, fast, and theoretically-grounded** method for synthesizing multimodal outliers. The key contributions that we believe will benefit the community are:

**Simplicity and Efficiency**: A straightforward, modality-agnostic approach that achieves state-of-the-art performance with a 10x to 370x speedup over previous methods, making effective multimodal outlier synthesis more accessible.

**Broad Applicability**: As demonstrated in our rebuttal, Feature Mixing is effective not only across diverse two-modality settings (e.g., LiDAR-image, video-flow) but also extends seamlessly to three-modality and even single-modality scenarios.

**Rigorous Evaluation**: We introduce CARLA-OOD, a new public benchmark for multimodal OOD segmentation that enables cleaner and more realistic evaluation, addressing limitations in existing datasets.

The reviewers' feedback was important in helping us better highlight these contributions and the versatility of our method. We thank the reviewers and ACs once again for their time and constructive engagement throughout this process.

---

### Decision · Program_Chairs · 2025-09-17

**Decision:**

Accept (poster)

**Comment:**

This paper proposes an outlier synthesis technique for multi-modal OOD detection based on the simple yet effective idea of feature mixing. The authors successfully addressed the reviewers' concerns during the discussion phase, and as a result, all the reviewers unanimously support acceptance. The strengths highlighted by the reviewers are: strong empirical results, simplicity of the proposed method, flexibility across diverse multi-modal settings, the introduction of a new multi-modal OOD benchmark, and clear presentation.

Therefore, AC recommends acceptance of this submission. The authors are strongly encouraged to incorporate the reviewers' feedback (e.g., additional experimental results, extended discussions, clarifications, and implementation details), as doing so would further strengthen and clarify the contributions of this work.